

# Exactly solvable lattice Hamiltonians and gravitational anomalies

Yu-An Chen[1,2] and Po-Shen Hsin[1,3]

**1** Walter Burke Institute for Theoretical Physics, California Institute of Technology,
Pasadena, CA 91125, USA
**2** Department of Physics, Condensed Matter Theory Center, and Joint Quantum Institute,
University of Maryland, College Park, Maryland 20742, USA
**3** Mani L. Bhaumik Institute for Theoretical Physics, 475 Portola Plaza,
Los Angeles, CA 90095, USA

## Abstract

We construct infinitely many new exactly solvable local commuting projector lattice Hamiltonian models for general bosonic beyond group cohomology invertible topological phases of order two and four in any spacetime dimensions, whose boundaries are characterized by gravitational anomalies. Examples include the beyond group cohomology invertible phase without symmetry in (4+1)D that has an anomalous boundary $\mathbb{Z}_2$ topological order with fermionic particle and fermionic loop excitations that have mutual $\pi$ statistics. We argue that this construction gives a new non-trivial quantum cellular automaton (QCA) in (4+1)D of order two. We also present an explicit construction of gapped symmetric boundary state for the bosonic beyond group cohomology invertible phase with unitary $\mathbb{Z}_2$ symmetry in (4+1)D. We discuss new quantum phase transitions protected by different invertible phases across the transitions.

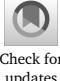

# 1   Introduction

Invertible phases of matter [1–5] are gapped phases with a unique vacuum, and they have applications such as topological insulators or superconductors [6] and are useful in understanding the relation between symmetry and phases of matter. For instance, phase transitions separated by invertible phases will be robust against perturbations; similarly, the interfaces that separate different invertible phases are also stable against perturbations. A large class of invertible phases of matter with symmetry is described by the cohomology of the symmetry group [7]. These phases have lattice model realization. They can be understood by gauging the symmetry, which results in Dijkgraaf-Witten gauge theories [8], and different invertible phases can be distinguished from the topological property of non-local operators in the theory, such as topological spin and braiding (see *e.g.* Ref. [9]).

On the other hand, not every invertible phase can be described by the cohomology of the symmetry group, such as invertible phases without symmetry. While the phases transitions separating different invertible phases with symmetry might not be robust against symmetry breaking perturbations, the phase transition between different invertible phases without symmetry is always robust. These phases that are not described by group cohomology are known

as beyond group cohomology invertible phases, and are classified in Refs. [7, 8]. While the interfaces between different group cohomology invertible phases are protected by the 't Hooft anomaly of the unitary symmetry, the interfaces between different beyond group cohomology invertible phases are protected by both the 't Hooft anomaly of the unitary symmetry and certain gravitational anomaly [10, 11].

However, most condensed matter systems do not have exact Poincaré symmetry, and it is difficult to study condensed matter systems on general curved spacetime. Thus to better understand these beyond group cohomology invertible phases, it is desirable to describe such phases using lattice Hamiltonians. The lattice model also provides a construction for the "anomalous" boundary state for the invertible phase. For instance, the boundaries of invertible phases without symmetry are expected to have a gravitational anomaly, and it might be interesting to investigate how the gravitational anomaly, which relies on the emergent Poincaré symmetry at low energy, manifests itself on the boundary of the lattice model. By modifying the lattice model, one can also study possibly continuous quantum phase transitions between different invertible phases.

Unlike the "group cohomology phases", exactly solvable lattice models for beyond group cohomology invertible phases have not been constructed in generality. An example of such lattice model for the beyond group invertible phase with $\mathbb{Z}_2$ unitary symmetry in (4+1)D is constructed in Ref. [12] using techniques from quantum cellular automata (QCA). See also Ref. [13] for previous attempts of understanding such phases. However, such a lattice model is not readily generalized to other beyond group cohomology invertible phases.

In this note we develop a method to construct exactly solvable Hamiltonian models for general bosonic beyond group cohomology invertible phases, in any spacetime dimensions. The models have the following features: (1) the Hamiltonian is a sum of local commuting projectors and thus is exactly solvable, (2) the model can be defined on any triangulated lattice or hypercubic lattice, (3) the dimension of the local Hilbert space is finite. We will focus on the phases without time-reversal symmetry. Such invertible phases are classified by cobordism groups $\Omega_{SO}^{d+1}(BG)$ for unitary ordinary symmetry $G$ [14–17]. We will focus on a particular infinite subset of bosonic invertible phases with unitary $G$ symmetry. These phases have order 2 or order 4 under stacking: two copies or four copies of the phases stacked on top of each other can be smoothly connected to the trivial phase.[1] Examples of such phases are discussed in Refs. [14, 15].

Some invertible phases can be expressed using invertible gauge theory, *i.e.* gapped gauge theory with a unique ground state on any space.[2] If we replace the dynamical gauge field with background gauge field, we obtain the symmetry protected topological (SPT) phase with effective action given by the topological action of the gauge field, and the invertible gauge theory is obtained from the SPT phase by gauging the symmetry, *i.e.* promoting the background gauge field to be dynamical. Such invertible gauge theory has no reference to Poincaré symmetry and in principle can be defined on the lattice. In particular, the invertible gauge theory obtained by gauging symmetry in group cohomology invertible phases can be described by exactly solvable lattice models. For instance, the invertible phases with $\mathbb{Z}_2$ fermion parity symmetry in (1+1)D can be described by dynamical $\mathbb{Z}_2$ gauge theory with certain topological action [22].

---

[1]In other words, this excludes the "chiral" bosonic beyond group cohomology invertible phases whose effective actions depend on the Pontryagin classes $p_i(TM)$ but not through $p_i(TM)$ mod 4 (which can be expressed in terms of the (higher-dimension) Pontryagin square of the Stiefel-Whitney classes [18]). An example of beyond group cohomology SPT phase not discussed here is the SPT phase in (4+1)D with $\mathbb{Z}_3$ symmetry and the effective action $\frac{2\pi}{3} \int A_{\mathbb{Z}_3} \cup p_1(TM)$, where $A_{\mathbb{Z}_3}$ is the background $\mathbb{Z}_3$ gauge field.

[2]When the gauge field has fluxes, the topological action of the gauge field in invertible gauge theories attaches the flux to electric excitation in one dimension higher by the generalization of the Witten effect, such that the flux confines and the theory is invertible. An example is the $\mathbb{Z}_2$ two-form gauge theory in 3+1d with minimal topological action, which is invertible because the flux particle is attached to an electric string and becomes confined (see *e.g.* Ref. [19–21]).

Similarly, the invertible phases with fermion parity symmetry in (2+1)D can be described by Chern-Simons theory with gauge group $SO(N)$ for integer $N$ at Chern-Simons level one (see *e.g.* Ref. [23, 24]).[3]

In this note, we show that an infinite class of the bosonic beyond group cohomology invertible phases can be described by invertible gauge theory: this includes all bosonic beyond group cohomology invertible phases of order two and four in any spacetime dimension. Such theory is obtained by gauging the symmetry in a "parent" group cohomology invertible phase. Since the group cohomology invertible phases have exactly solvable lattice Hamiltonian model, this provides a systematic lattice Hamiltonian model construction for the beyond group cohomology invertible phases (at least for those of order two or four), by gauging the symmetry in the exactly solvable lattice model for the corresponding "parent" group cohomology invertible phase. The procedure of constructing the lattice model for the invertible phases is

(1) Construction of an exactly solvable Hamiltonian for the "parent" group cohomology invertible phase.

(2) Gauging the symmetry (or a subgroup of the symmetry) in the above lattice model to obtain the invertible gauge theory that describes the beyond group cohomology invertible phase.[4]

(3) We obtain an "anomalous" boundary state by truncating the lattice model near the boundary.

(4) We construct lattice Hamiltonians for new quantum phase transitions protected by "gravitational response" described by different invertible phases. This is the analogue of free fermion critical point in (2+1)D protected by the change in the thermal Hall response across the transition.

In Section 2 we will discuss the above procedure in detail. Examples of such constructions have been discussed in Refs. [21, 26].

The non-triviality of the invertible phase described by the bulk lattice model can be inferred from the anomalous boundary topological order. In the main example considered in this work, the boundary coincides with the known anomalous $\mathbb{Z}_2$ topological order [27], which implies the bulk is a non-trivial invertible phase.

The gravitational anomaly on the boundary often manifests as non-trivial self-statistics and mutual-statistics of excitations. For instance, if an excitation has non-trivial self-statistics, it requires a framing to be defined on a curved spacetime. If it has mutual-statistics with another excitation, the framing cannot be extended to the other excitation.[5] If the other excitation also has non-trivial self-statistics and itself requires a framing to be defined on the curved spacetime, this will lead to inconsistency in defining both excitations on the curved spacetime since no framing can be defined for both excitations. This is a gravitational anomaly, and it is a global

---

[3]More recently, the invertible phases in (3+1)D with one-form symmetry and time-reversal symmetry that forms a "mixture" two-group is constructed using dynamical $\mathbb{Z}_2$ two-form gauge theory with certain topological action in Ref. [21].

[4]In the Hamiltonian models, the Gauss law constraints are imposed energetically rather than exactly, and thus the Hilbert space still has a tensor product structure. For instance, the $\mathbb{Z}_2$ toric code model in 2+1d [25] describes $\mathbb{Z}_2$ gauge theory only at the low energy subspace.

[5]For instance, the expectation value of the line operator creating fermion is given by the local spin structure on the line, which is a geometric object. If the line operator braids with another operator, the holonomy of the local spin structure is a $c$-number multiple of the identity operator and cannot reproduce the braiding, and thus the local spin structure cannot be defined near the other operator with which the line operator braids non-trivially. We note that in this case, if the theory also has a local fermionic particle, then we can dress the line operator with the local fermion to make it a boson, which no longer requires a framing, and the anomaly becomes trivial.

anomaly since statistics is non-local. We will see such phenomena explicitly on the boundary of the lattice models we constructed for invertible phases.

The work is organized as follows. In Section 2, we describe the general procedure of constructing local commuting projector lattice models for general beyond group cohomology invertible phases of order two and four in any spacetime dimension. In Section 3, we illustrate the construction using the example of invertible phase in (4+1)D without symmetry. In Section 4, we apply the method to construct a new lattice Hamiltonian model for the beyond group cohomology invertible phase with unitary $\mathbb{Z}_2$ symmetry in (4+1)D. In Section 5, we discuss the results and some future directions.

There are several appendices. In Appendix A, we review the mathematical properties of cochains and cup products used in the construction of lattice Hamiltonians. In Appendix B, we discuss another approach to obtain boundary state in the Hamiltonian model by gauging a symmetry both in the bulk and on the boundary. In Appendix C, we use (2+1)D toric code model to illustrate the method of obtaining boundary theory by truncating the bulk Hamiltonian. In Appendix D, we give the details for the computation of the boundary Hamiltonian of the invertible phase without symmetry in (4+1)D.

## 1.1 Summary of main examples

We use two examples to illustrate our construction for lattice Hamiltonian models of bosonic beyond group cohomology invertible phases of order two and four in the general spacetime dimension. The examples are in (4+1)D spacetime. The first example does not have symmetry, and the second example has $\mathbb{Z}_2$ unitary symmetry. Both phases have order two.

### 1.1.1 Bosonic invertible phase without symmetry in (4+1)D

The (4+1)D beyond group cohomology SPT phases without symmetry have $\mathbb{Z}_2$ classification [14], and the non-trivial phase is characterized by the effective action

$$\pi \int w_2(TM) \cup w_3(TM) = \pi \int w_2(TM) \cup Sq^1 w_2(TM), \tag{1}$$

where we consider orientable spacetime manifolds.

This beyond group cohomology invertible phase has several known boundary states in (3+1)D, including the all-fermion electrodynamics with heavy fermionic electric particles and monopoles [28–32], and the anomalous $\mathbb{Z}_2$ topological order with fermionic particle and fermionic loop excitations [30,32,33], and the $SU(2)$ gauge theory with odd number of massless fermion with isospin 3/2 [34]. All these boundary theories have fermionic loops and a fermionic particle: they have fermion Wilson line due to the spin/charge relation, and also have $\pi$ flux monopole whose Dirac string is a fermionic string, since the $2\pi$ flux monopole is a fermion.[6]

We describe the beyond group cohomology invertible phase using the invertible gauge theory with $\mathbb{Z}_2$ two-form $a_2$ and three-form $b_3$ gauge fields, and the action

$$\pi \int a_2 \cup Sq^1(a_2) + \pi \int Sq^2(b_3) + \pi \int a_2 \cup b_3. \tag{2}$$

The local commuting projector Hamiltonian for the invertible phase (1) described by the above invertible gauge theory with dynamical gauge fields $a_2, b_3$ is as follows. We put $\mathbb{Z}_2$ qubits on

---

[6]This follow from the property that the $2\pi$ flux monopole is a fermion and thus attached to $\int w_2(TM)$, while the Dirac string of the $\pi$ flux monopole is attached to $\int dw_2(TM)/2 = \int w_3(TM)$, which implies it is a fermionic string (the property of fermionic string is that it is attached to $\int w_3(TM)$ [30]).

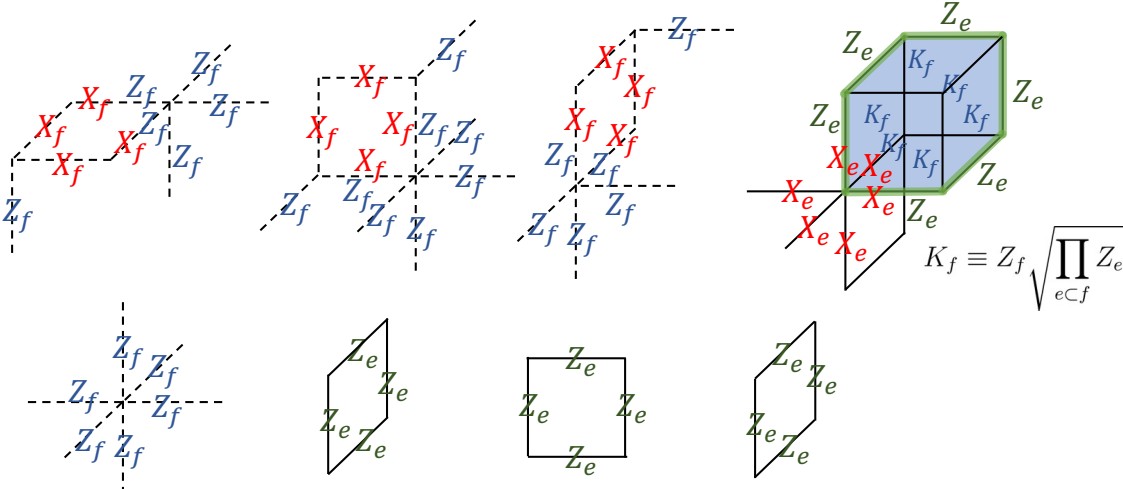

Figure 1: The Hamiltonian for the boundary state obtained by truncating the Hamiltonian model of the beyond group cohomology invertible phase without symmetry in (4+1)D. There are qubits on the edges and faces of the lattice, acted on by the Pauli operators $X_e, Y_e, Z_e$ and $X_f, Y_f, Z_f$, respectively. Solid lines in the figure are edges in the cubic lattice, while dashed lines are edges in the dual lattice, *i.e.* faces in the original lattice. The first row imposes the Gauss law energetically, and the second row imposes the zero flux condition energetically, which are violated by the excitations in Fig. 2. In the figure we omit the bulk terms. In the figure, $K_f = Z_f \sqrt{Z_{e_1} Z_{e_2} Z_{e_3} Z_{e_4}}$ for the edges $e_1, e_2, e_3, e_4$ that bound the face $f$, and the square root takes value in $\{1, i\}$ in the eigenbasis of $Z_{e_1} Z_{e_2} Z_{e_3} Z_{e_4}$.

each two- and three-simplices, acted by Pauli matrices $X_f, Z_f$ and $X_t, Z_t$.

$$
\begin{aligned}
H_{w_2 w_3} = &-\sum_f \prod_{t \supset f} X_t \prod_{t'} Z_{t'}^{\int t' \cup_2 \delta f} \prod_{f'} Z_{f'}^{\int f' \cup f} \\
&- \sum_e \prod_{f \supset e} X_f \prod_{f'} Z_{f'}^{\int f' \cup (e \cup_1 \delta e)} \prod_t Z_t^{\int e \cup t} \prod_{f_1, f_2} CZ(Z_{f_1}, Z_{f_2})^{\int e \cup (f_1 \cup_1 f_2) + f_1 \cup (f_2 \cup_2 \delta e)} \\
&- \sum_t \prod_{f \subset t} Z_f - \sum_{\text{4-simplex } p} \prod_{t \subset p} Z_t,
\end{aligned}
\tag{3}
$$

where $e$ denotes the 1-cochain that takes value 1 on the edge $e$ and zero on all other edges, and similarly for 2-cochain $f$ and 3-cochain $t$, and we introduced

$$
CZ(a, b) = \begin{cases} -1, & \text{if } (a, b) = (-1, -1), \\ 1, & \text{if } (a, b) = (1, 1), (1, -1), (-1, 1). \end{cases}
\tag{4}
$$

In the above formula, $CZ(Z_i, Z_j)$ is the controlled-Z gate.

**Boundary state**   By truncating the above bulk Hamiltonian on the boundary (Fig. 1), we find fermionic particle and fermionic loop excitations (Fig. 2) on the boundary with $\pi$ statistics, in agreement with the expected anomalous boundary state for the beyond group cohomology SPT phase (1) as discussed in Refs. [30, 33]. The fermionic loop excitation is characterized by a T-junction process that involves orientation reversal (for more detail, see Section 3.4). In an upcoming work [35] we show the statistics of the fermionic loop excitation can also be described by a double commutator of the loop creation operator with a suitable choice of a

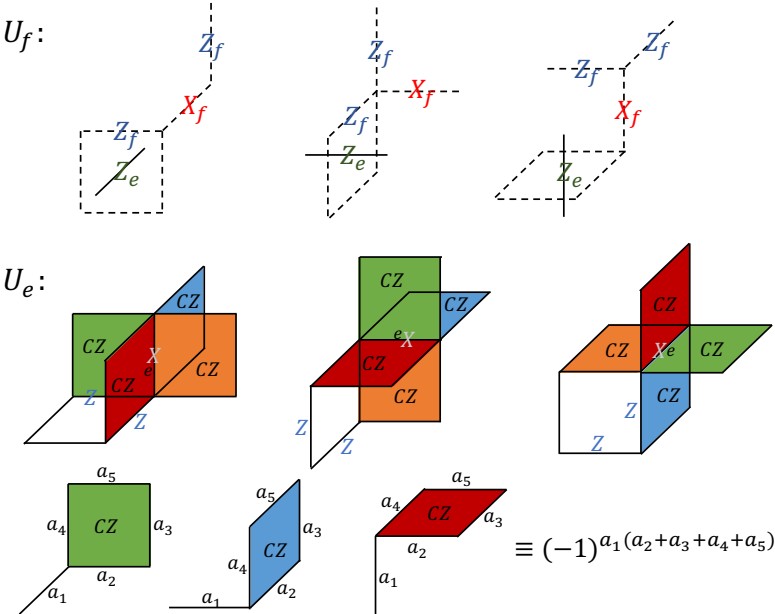

Figure 2: The excitations in the boundary state of the beyond group cohomology invertible phase without symmetry in (4+1)D.

branching structure, similar to the statistics of fermionic particles described by the commutator of the fermion hopping operator.

We remark that such anomalous boundary state can also be obtained from the Higgs mechanism in the $U(1)$ Maxwell theory with a Higgs scalar with the charge of two, and where both the unit electric particle and the $2\pi$-flux magnetic monopole are fermions. There are fermionic loops from the Dirac string of the $\pi$ flux monopole and the analogue of the Dirac string for the half charge electric particle and dyon [35].

If we take two copies of the system, we can condense the boson made out of the fermion in each copy and make the boundary completely trivial. Thus the invertible phase has order two.

### 1.1.2 Bosonic invertible phase with $\mathbb{Z}_2$ unitary symmetry in (4+1)D

Another example we use to illustrate our construction is the beyond group cohomology invertible phase with $\mathbb{Z}_2$ unitary symmetry in (4+1)D, with the effective action

$$\pi \int \mathcal{C}_1 \cup Sq^2(w_2), \tag{5}$$

where $\mathcal{C}_1$ is the $\mathbb{Z}_2$ background gauge field for the $\mathbb{Z}_2$ symmetry. The theory can be described by an invertible gauge theory with $\mathbb{Z}_2$ two-form $a_2$ and three-form $b_3$ gauge fields, and the action

$$\pi \int \mathcal{C}_1 \cup Sq^2(a_2) + \pi \int Sq^2(b_3) + a_2 \cup b_3. \tag{6}$$

The Hamiltonian model for the beyond group cohomology SPT phase as described by the above invertible gauge theory is as follows. We consider Euclidean lattice, and qubit on each

vertex, face and cube:

$$H_{Cw_2^2} = -\sum_v X_v \prod_{f_1,f_2} CZ(Z_{f_1}, Z_{f_2})^{\int v \cup f_1 \cup f_2} - \sum_e \prod_{f \supset e} X_f \prod_t Z_t^{\int e \cup t} \prod_{v,f'} CZ(Z_v, Z_{f'})^{\int \delta v \cup (f' \cup_1 \delta e)}$$
$$- \sum_f \prod_{t \supset f} X_t \prod_t Z_t^{\int t \cup_2 \delta f} \prod_{f'} Z_{f'}^{\int f' \cup f} - \sum_t \prod_{f \subset t} Z_f - \sum_{\text{4-simplex } p} \prod_{t \subset p} Z_t \,. \tag{7}$$

We remark that a different lattice Hamiltonian model for the same beyond group cohomology invertible phase in (4+1)D was constructed in Ref. [12].

**Boundary state**    The boundary state of the theory can be obtained by truncating the bulk Hamiltonian, and is given by an anomalous $\mathbb{Z}_2$ topological order with fermionic particle and bosonic loop excitations, but the $\mathbb{Z}_2$ symmetry realized anomalously on the loop excitation. To be specific, the intersection of the loop excitation with the domain wall implementing the $\mathbb{Z}_2$ symmetry has the statistics of a fermionic particle. A related symmetric gapped boundary is also discussed in Ref. [12].

If we take two copies of the system, we can condense the boson particle made out of the fermion in each copy and make the boundary completely trivial. Thus the invertible phase has order two.

# 2    General construction of exactly solvable Hamiltonian models

## 2.1    Beyond group cohomology invertible phases from "parent" group cohomology invertible phases

We will show that the bosonic invertible phases of order two and four can be obtained by gauging certain $\mathbb{Z}_2$ higher-form symmetry in a group cohomology invertible phase. Mathematically, these invertible phases correspond to the elements of order 2 or 4 in

$$H^{d+1}(BG \times BSO(d+1), U(1))/\{v_k = 0 \text{ for } k > (d+1)/2\}\,, \tag{8}$$

where $SO(d+1)$ is the Lorentz group for the $(d+1)$-dimensional spacetime with Euclidean signature, $v_k$ is the $k$th Wu class [36] on $BSO(d+1)$. A useful way to describe invertible phases is using their low energy effective action in the presence of background probe gauge field for the symmetry, and the elements in (8) corresponds to such effective action as follows. Let us denote the background gauge field collectively by $A$ (it can be the background gauge field for internal symmetry, or some probe curved gravity background). It is a map from the spacetime to $BG \times BSO(d+1)$: for instance, it assigns a $G$ element to each curve on the spacetime. Then for element $\omega$ in (8), the effective action for the background gauge field is given by the pullback $A^*\omega$ of $\omega$ by $A$.

We use the property that the bosonic beyond group cohomology SPT phases with finite group $G$ unitary symmetry have effective actions described in turns of the background gauge field for the $G$ symmetry and the Stiefel-Whitney classes $w_i(TM)$, Pontryagin classes $p_i(TM)$ for the tangent bundle, as well as the higher-dimensional generalization of gravitational Chern-Simons terms [14–16]. For the invertible phases of order two and four, the effective action for the invertible phases only depend on the $G$ background gauge field, and the $\mathbb{Z}_2$ valued Stiefel-Whitney classes for the tangent bundle.

The Stiefel-Whitney classes can be described by dynamical higher-form $\mathbb{Z}_2$ gauge field in invertible gauge theories. In $d+1$ spacetime dimension, an $n$-form $\mathbb{Z}_2$ gauge field $b_n$ satisfies the identity [36]

$$Sq^{d+1-n}(b_n) = v_{d+1-n}(TM) \cup b_n\,, \tag{9}$$

where $v_r(TM)$ is the $r$th Wu class of the tangent bundle $TM$. Then we can express the Wu class using another $\mathbb{Z}_2$ $(d+1-n)$-form gauge field $a_{d+1-n}$ that couples to $b_n$, with the action

$$\pi \int Sq^{d+1-n}(b_n) + a_{d+1-n} \cup b_n = \pi \int (v_{d+1-n}(TM) + a_{d+1-n}) \cup b_n \bmod 2\pi\mathbb{Z}. \tag{10}$$

Then integrating out the Lagrangian multiplier $b_n$ implies

$$a_{d+1-n} = v_{d+1-n}(TM), \tag{11}$$

and thus the fluctuation of the dynamical gauge fields is suppressed, and the theory is invertible.[7] Thus we can express the $r$th Wu class for $r \leq d+1$ using the $\mathbb{Z}_2$ gauge fields $a_r$. Note the Wu classes are non-trivial for $r \leq (d+1)/2$. The Stiefel-Whitney classes can be obtained from the Wu classes as [36]

$$w_r(TM) = \sum_{j=\max(0,r-[(d+1)/2])}^{[r/2]} Sq^j(v_{r-j}(TM)) = \sum_{j=\max(0,r-[(d+1)/2])}^{[r/2]} Sq^j(a_{r-j}), \tag{12}$$

where the upper and lower bounds in the summation are imposed to exclude indices with zero summands. Thus we can express the bosonic beyond group cohomology phases as an invertible gauge theory for the $n$-form $\mathbb{Z}_2$ gauge fields $b_n$ and $(d+1-n)$-form $\mathbb{Z}_2$ gauge fields $a_{d+1-n}$ for $[(d+1)/2] \leq n \leq d$, in addition to the $G$ background gauge field, with action described by the group cohomology

$$H^{d+1}\left(BG \times \bigotimes_{n=[(d+1)/2]}^{d} \left(B^{n+1}\mathbb{Z}_2^{(b_n)} \times B^{d+1-n}\mathbb{Z}_2^{(a_{d+1-n})}\right), U(1)\right). \tag{13}$$

Thus, the Hamiltonian model for bosonic beyond group cohomology SPT phases can be constructed by the following steps.

(1) First, construct the local commuting projector Hamiltonian model for the "parent" SPT phase with 0-form symmetry $G$, $(n-1)$-form and $(d-n)$-form $\mathbb{Z}_2$ symmetries with $[(d+1)/2] \leq n \leq d$, described by the group cohomology (13). The group cocycle has the form

$$\omega(g, \{(a_r, b_r)\}) = \omega_0(g, \{a_r\}) \prod_{n=[(d+1)/2]}^{d} (-1)^{\int Sq^{d+1-n}(b_n) + a_{d+1-n} \cup b_n}, \tag{14}$$

where $g \in G$. We will discuss the construction in Section 2.2.

(2) Then, gauge the higher-form symmetry with gauge fields $a_{d+1-n}, b_n$. This then produces a local commuting projector Hamiltonian model for the invertible gauge theory that describes the beyond group cohomology SPT phase, since the $\mathbb{Z}_2$ $n$-form gauge field $b_n$ is the Lagrangian multiplier that enforces $a_{d+1-n} = v_{d+1-n}(TM)$. We will discuss the explicit construction in Section 2.3.

(3) We obtain a boundary state of the beyond group cohomology SPT phase by truncating the terms in the bulk Hamiltonian near the boundary. We will discuss in Section 2.4.

---

[7]In particular, if the above two terms are the only topological action, then the partition function on any closed manifold equals one.

(4) We add "transverse field term" $\sum_s Z_s$ to the lattice model, where $Z_s$ is the Pauli $Z$ matrix acting on the qubit on simplex $s$, to obtain a lattice model that is not a sum of commuting projectors. If the new term has a large coefficient, it derives the new model to the trivial phase, and at some finite coefficient it describes phase transitions protected by the invertible phase.

We remark that a related but different construction is discussed in [37], which obtains a gapped boundary for the bosonic beyond group cohomology SPT phases by introducing a dynamical $\mathbb{Z}_2$ gauge field on the boundary that obeys twisted cocycle condition given by the Stiefel-Whitney classes of the tangent bundle. This allows one to rewrite the bulk effective action for the beyond group cohomology SPT phases as a topological boundary term for these twisted dynamical boundary gauge fields. In our construction for the bulk invertible phases, we introduce instead a bulk dynamical gauge field that is constrained dynamically to give the Stiefel-Whitney classes of the tangent bundle. If there is a boundary, this allows us to discuss different boundary conditions for such dynamical bulk gauge fields. We will focus on the rough or Dirichlet boundary condition, and also briefly discussed the free boundary condition. The gapped boundary in [37] corresponds to the Dirichlet boundary condition which will also be discussed in our work.

## 2.2 Hamiltonian model for "parent" group cohomology SPT phases

Let us review a lattice model construction of group cohomology SPT phase with $G$ 0-form symmetry and $\mathbb{Z}_2$ $n_i$-form symmetries for collection of $\{n_i\}$. The lattice is $\mathbb{Z}^d$ for space dimension $d$, or any triangulated manifold. The discussion follows the approach in Refs. [9, 23, 38–40]. The Hilbert space is labelled by $g(v) \in G$ at each vertex $v$, $\lambda^{(n_i)}(s_{n_i}) \in \mathbb{Z}_2$ at each $n_i$-simplex $s_{n_i}$. The SPT phase can be describe by $d+1$-dimensional topological action $\omega_{d+1}(x, \{c^{(n_i+1)}\})$ for background one-form $x$ for $G$ symmetry and $(n_i+1)$-form $c^{(n_i+1)}$ for the $\mathbb{Z}_2$ $n_i$-form symmetries. We have

$$\omega_{d+1}(x^g, \{c^{(n_i+1)} + \delta\lambda^{(n_i)}\}) - \omega_{d+1}(x, \{c^{(n_i+1)}\}) = \delta\omega'_d(x, \{c^{(n_i+1)}\}; g, \{\lambda^{(n_i)}\}), \qquad (15)$$

where $\omega'_d$ has degree $d$, and $x^g$ is the gauge transformation of $x$ by $g$, i.e., $x^g(e_{ij}) = g(v_i)^{-1}x(e_{ij})g(v_j)$. Denote $\omega'_d$ with $x, c^{(n_i+1)} = 0$ as $\omega'^0_d$.

We would like to construct a Hamiltonian with a unique ground state described by the bulk wavefunction

$$|\Psi\rangle = \sum_{g,\{\lambda^{(n_i)}\} \text{ in bulk}} e^{i \int_{\text{space}} \omega'^0_d(g, \lambda^{(n_i)})} |g, \{\lambda^{n_i}\}\rangle, \qquad (16)$$

where in the summand, $\omega'_d(g, \lambda^{(n_i)})$ is replaced by its eigenvalues on the eigenstate $|g, \{\lambda^{n_i}\}\rangle$. The summation is over all the configurations in the bulk. If the space has a boundary, the wavefunction depends on the boundary configurations, which are not summed over.

A Hamiltonian for the SPT phase wavefunction can be obained by conjugating the paramagnet $H^0 = -\sum_v \sum_{h \in G} X^h_v - \sum_{s_n} X_{s_n}$ by $e^{i \int \omega'^0_d(g, \lambda^{(n_i)})}$:

$$H_{\text{SPT}} = -\sum_v \sum_{h \in G} X^h_v e^{i \int \omega'^0_d(\rho_{h^v} g, \{\lambda^{(n_i)}\}) - i \int \omega'^0_d(g, \{\lambda^{(n_i)}\})} - \sum_{s_{n_i}} X_{s_{n_i}} e^{i \int \omega'^0_d(g, \{\lambda^{(n_i)} + s_{n_i}\}) - i \int \omega'^0_d(g, \{\lambda^{(n_i)}\})}, \quad (17)$$

where $\rho_{h^v} g$ change the 0-cochain $g(v)$ by left multiplication with $h$ for the 0-cochain at vertex $v$ (as implemented by the operator $X^h_v$), and $s_{n_i}$ is the $n_i$-cochain with value 1 on the $n_i$-simplex $s_{n_i}$ and 0 elsewhere. [8] This gives a local commuting projector Hamiltonian. Since the paramagnet Hamiltonian $H^0$ has the ground state given by superposition of all the configurations with equal weight, the Hamiltonian $H_{\text{SPT}} = e^{i \int \omega'^0_d} H^0 e^{-i \int \omega'^0_d}$ has ground state given by $|\Psi\rangle$.

---

[8] $\lambda^{(n_i)} + s_{n_i}$ modifies the $\mathbb{Z}_2$ variable of $\lambda^{(n_i)}$ at the single $n_i$-simplex $s_{n_i}$.

## 2.3 Gauging a symmetry in Hamiltonian model

Let us gauge the $n_i$-form symmetry in the above Hamiltonian. The discussion follows the approach in Refs. [9, 23, 39–42]. We introduce qubits on each $(n_i + 1)$-simplex, acted by Pauli matrices $\overline{X}_{s_{n_i+1}}, \overline{Y}_{s_{n_i+1}}, \overline{Z}_{s_{n_i+1}}$, and we denote $c^{(n_i+1)}_{s_{n_i+1}} = (1 - Z_{s_{n_i+1}})/2$.

- We demand the new gauge fields $c^{(n_i+1)}$ are flat in low energy subspace. Thus we add flux terms in the Hamiltonian to penalize the configuration with $\delta c^{(n_i+1)} \neq 0 \bmod 2$

$$H_{\text{flux}} = -\sum_{s_{n_i+2}} \prod \overline{Z}_{s_{n_i+1}} = -\sum_{s_{n_i+2}} (-1)^{\int_{s_{n_i+2}} \delta c^{(n_i+1)}}, \tag{18}$$

  where the product is over all $(n_i + 1)$-simplices $s_{n_i+1}$ on the boundary of the $(n_i + 2)$-simplex $s_{n_i+2}$.

- We impose the Gauss law constraint

$$X_{s_{n_i}} \prod \overline{X}_{s_{n_i+1}} = 1, \tag{19}$$

  where the product is over all $(n_i + 1)$-simplices $s_{n_i+1}$ adjacent to the $n_i$-simplex $s_{n_i}$. We note the Gauss law constraint commutes with the flux terms $H_{\text{flux}}$.

- For the Hamiltonian to commute with the Gauss law constraint, the original Hamiltonian must be modified to be invariant under $\lambda^{(n_i)} \to \lambda^{(n_i)} + s'_{n_i}$, $c^{(n_i+1)} \to c^{(n_i+1)} + \delta s'_{n_i}$. The modification follows from the cocycle $\omega_{d+1}$ with nonzero $c^{(n_i+1)}$ as

$$\begin{aligned}
&\delta\omega'_d(x^g, \{c^{(n_i+1)}\}; g, \lambda^{(n_i)} + s'_{n_i}) - \delta\omega'_d(x^g, \{c^{(n_i+1)}\}; g, \lambda^{(n_i)}) \\
=&\left(\omega_{d+1}(x^g, \{c^{(n_i+1)} + \delta\lambda^{(n_i)} + \delta s'_{n_i}\}) - \omega_{d+1}(x^g, \{c^{(n_i+1)}\})\right) \\
&-\left(\omega_{d+1}(x^g, \{c^{(n_i+1)} + \delta\lambda^{(n_i)}\}) - \omega_{d+1}(x^g, \{c^{(n_i+1)}\})\right) \\
=&\delta\omega'_d(x^g, \{c^{(n_i+1)} + \delta\lambda^{(n_i)}\}; g, s'_{n_i}).
\end{aligned} \tag{20}$$

  This gives a modification that is invariant under the transformation, since it depends on $c^{(n_i+1)}, \lambda^{(n_i)}$ by the invariant combination $c^{(n_i+1)} + \delta\lambda^{(n_i)}$.

  In some cases, we also need to conjugate the Hamiltonian by the projector to the zero flux sector of the introduced gauge fields to ensure the Hamiltonian is a sum of commuting terms.

- We can use Gauss law constraint to perform gauge-fixing to set $\lambda^{(n_i)} = 0$ and replace $X_{s_{n_i}}$ by $\prod \overline{X}_{s_{n_i+1}}$, where the product is over the $(n_i + 1)$-simplices $s_{n_i+1}$ adjacent to the $n_i$-simplex $s_{n_i}$.

This gives a local commuting projector Hamiltonian.

## 2.4 Boundary state from truncation of the bulk Hamiltonian

Let us discuss the boundary state of a bulk Hamiltonian. They can be obtained by looking at the bulk terms near the boundary. In particular, we can also deduce operators on the boundary and thus the boundary excitations. Examples of the truncating procedure for obtaining boundary Hamiltonian are discussed in *e.g.* Refs. [40, 43, 44]. In this paper, we follow the boundary construction in Refs. [26, 42].

Consider space $M_d$ with boundary $N_{d-1} = \partial M_d$. We construct an auxiliary closed manifold by introducing auxiliary simplices as follows. We first introducing a vertex $v_0$, and then

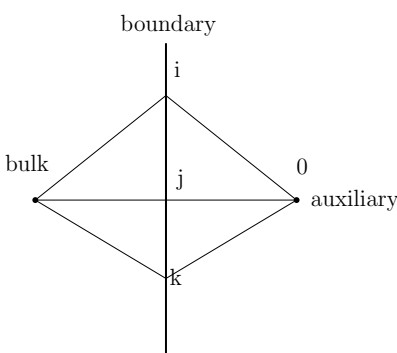

Figure 3: Auxiliary simplices connected to auxiliary vertex 0.

introduce auxiliary simplices by connecting $v_0$ to all vertices on the boundary $N_{d-1} = \partial M_d$, forming auxiliary edges $\langle 0i \rangle$, auxiliary faces $\langle 0ij \rangle$, with vertices $i, j, \cdots$ in $N_3$, *etc.* See Figure 3. The space formed by these auxiliary simplices and $N_{d-1}$ is denoted as $CN_{d-1}$, *i.e.*, the cone over topological space $N_{d-1}$. Gluing $CN_{d-1}$ and $M_d$ gives a closed spatial 4-manifold:

$$\widetilde{M}_d \equiv M_d \sqcup CN_{d-1}, \quad \partial \widetilde{M}_d = 0. \tag{21}$$

For simpicity, in the following we will take the space to have trivial topology. The state on $\widetilde{M}_d$ is related to that on $M_d$ and its boundary as follows: for $n$-cochain $c_n$,

$$|c_n|_{\widetilde{M}_d}\rangle = |c_n; c_{n-1}^0|_{\partial M_d}\rangle, \tag{22}$$

where $c_{n-1}^0$ is the $(n-1)$-cochain on the boundary $N_{d-1} = \partial M_d$ such that

$$c_{n-1}^0(v_1 v_2 \cdots v_{n-1}) := c_n(v_0 v_1 \cdots v_{n-1}). \tag{23}$$

We will focus on configuration where $c_n$ is a $n$-form gauge theory, and obeys cocycle condition $\delta c_n = 0$. On space with trivial topology (or trivial topology near the boundary) such as $\mathbb{Z}^d$ we can then express $c_n = \delta \phi_{n-1}$. We will set $\phi_{n-1} = 0$ on all auxiliary $(n-1)$-simplices. To illustrate the procedure, in Appendix C we discuss the truncation for bulk $\mathbb{Z}_2$ toric code model in (2+1)D [45], whose gapped boundaries are discussed in Ref. [46].

There are different kinds of truncation we can perform. In the one case, we introduce new degrees of freedom on the auxiliary simplices. This is the analogue of the "rough" boundary condition [46], or the Dirichlet boundary condition for the $\mathbb{Z}_2$ higher-form gauge fields. We will focus on this boundary condition in the discussions. In another case, we do not introduce new degrees of freedom and just keep original Hamiltonian terms in the bulk. This is the analogue of the "smooth" boundary condition [46], or the free boundary condition for the $\mathbb{Z}_2$ higher form gauge fields.[9]

## 3 Beyond group cohomology SPT phase without symmetry in (4+1)D

In this section, we construct a gauge theory model for the invertible phase in (4+1)D with effective action $\pi \int w_2 \cup w_3$. The gauge theory is a $\mathbb{Z}_2$ one-form and $\mathbb{Z}_2$ two-form gauge theory, with non-trivial topological twist in the action.

---

[9]In this approach, we may need to impose extra boundary terms by hand. For example, Section III of Ref. [47] uses this approach and finds all commuting terms on the boundary of the three-fermion Walker-Wang model.

Let us start by considering the SPT phase for $\mathbb{Z}_2$ one-form and two-form symmetries with the following effective ation

$$\pi \int B_3 \cup_1 B_3 + A_2 \cup B_3 + A_2 \cup (A_2 \cup_1 A_2), \tag{24}$$

where $A, B$ are $\mathbb{Z}_2$ cocycles obey $\delta A_2 = 0, \delta B_3 = 0$. They are the background gauge fields for the one-form and two-form symmetries, respectively. The lattice Hamiltonian model will be constructed in Section 3.1.

Next, we will gauge the one-form and two-form symmetries. In other words, we sum over $A, B$. Using the Wu formula $B_3 \cup_1 B_3 = Sq^2 B_3 = w_2 \cup B_3$ we can convert the first two terms in (24) into $(w_2 + A_2) \cup B_3$, and summing over $B_3$ imposes $A_2 = w_2$, and via the last term in (24) and using $w_2 \cup_1 w_2 = w_3$, the action reproduces the effective action $\pi \int w_2 \cup w_3$ of the beyond group cohomology SPT phase. In Section 3.2 we will carry out this gauging procedure and obtain an exactly solvable Hamiltonian for the $w_2 w_3$ invertible phase, by gauging the one-form and two-form symmetries in the Hamiltonian model for the SPT phase in Section 3.1.

## 3.1 "Parent" group cohomology SPT phase

The wavefunction of the SPT phase can be constructed as follows, using similar method as in Ref. [40]. Denote $\phi_5(A, B) = B_3 \cup_1 B_3 + A_2 \cup B_3 + A_2 \cup (A_2 \cup_1 A_2)$,

$$\phi_5(A, B)\big|_{A=\delta a, B=\delta b} = \delta \phi_4(a, b), \quad \phi_4(a, b) = b \cup b + b \cup_1 \delta b + a \cup (\delta a \cup_1 \delta a + \delta b), \tag{25}$$

where $a \in C^1(M_4, \mathbb{Z}_2)$ and $b \in C^2(M_4, \mathbb{Z}_2)$ are 1-cochains and 2-cochains on the space ($\phi_5$ is defined on the spacetime, while $\phi_4$ is defined on the space). The bulk wavefunction is thus

$$|\Psi\rangle = \sum_{a,b} e^{i\pi \int \phi_4(a,b)} |a, b\rangle, \tag{26}$$

where we assign $\mathbb{Z}_2$ elements to each edge $e$ and face $f$ by $a(e), b(f)$, and the integral is over the space. The wavefunction can be written as

$$|\Psi\rangle = U |\Psi_0\rangle, \tag{27}$$

with $|\Psi_0\rangle = \sum_{a,b} |a, b\rangle$ being the ground state of the trivial Hamiltonian $H_0 = -\sum_e X_e - \sum_f X_f$, and the operator $U$ defined as

$$U |a, b\rangle \equiv (-1)^{\int \phi_4(a,b)} |a, b\rangle. \tag{28}$$

The Hamiltonian with $|\Psi\rangle$ as the ground state is $H = U H_0 U^\dagger$:

$$H = -\sum_e X_e (-1)^{i\pi \int \phi_4(a+e,b) - \phi_4(a,b)} - \sum_f X_f (-1)^{i\pi \int \phi_4(a,b+f) - \phi_4(a,b)}, \tag{29}$$

where $e$ (respectively, $f$) is the 1-cochain (respectively, 2-cochain) that takes value 1 on the edge $e$ (respectively, the face $f$), and 0 elsewhere. More explicitly,

$$\phi_4(a, b+f) - \phi_4(a, b) = \delta b \cup_2 \delta f + \delta a \cup f + \delta \phi_3(b, f),$$
$$\phi_3 = b \cup_1 f + \delta b \cup_2 f + a \cup f,$$
$$\phi_4(a+e, b) - \phi_4(a, b) = \delta a \cup (e \cup_1 \delta e + \delta a \cup_2 \delta e) + e \cup (\delta a \cup_1 \delta a + \delta b) + \delta \phi_3'(a, e),$$
$$\phi_3' = a \cup [(\delta a \cup_2 \delta e) + e \cup_1 \delta e]. \tag{30}$$

On closed space manifolds, the integral of $\delta\phi_3$ and $\delta\phi_3'$ can be dropped. One important feature is that on a closed space manifold, the Hamiltonian depends on $a, b$ only through $\delta a$ and $\delta b$. This is consistent with the one-form and two-form symmetry actions $a \to a + \lambda_1$ and $b + \lambda_2$ for any $\lambda_1 \in Z^1(M_4, \mathbb{Z}_2)$ and $\lambda_2 \in Z^2(M_4, \mathbb{Z}_2)$, i.e., $\delta\lambda_1 = 0$ and $\delta\lambda_2 = 0$. The bulk Hamiltonian can also be written as

$$
\begin{aligned}
H_{\text{bulk}} = & -\sum_f X_f \prod_t W_t^{\int t \cup_2 \delta f} \prod_{f'} W_{f'}^{\int f' \cup f} \\
& -\sum_e X_e \prod_f W_f^{\int f \cup (e \cup_1 \delta e)} \prod_t W_t^{\int e \cup t} \prod_{f_1, f_2} CZ(W_{f_1}, W_{f_2})^{\int e \cup (f_1 \cup_1 f_2) + f_1 \cup (f_2 \cup_2 \delta e)},
\end{aligned}
\tag{31}
$$

where we introduced

$$
W_t \equiv \prod_{f \subset t} Z_f, \quad W_f \equiv \prod_{e \subset f} Z_e, \quad CZ(i,j) = \begin{cases} -1, & \text{if } (i,j) = (-1,-1), \\ 1, & \text{if } (i,j) = (1,1), (1,-1), (-1,1). \end{cases}
\tag{32}
$$

For the boundary Hamiltonian, we consider a special case with $a \to a + \delta v$, where $v$ is a vertex on the boundary. Since

$$
\begin{aligned}
\int_{M_4} \phi_4(a + \delta v, b) - \phi_4(a, b) &= \int_{M_4} \delta v \cup \delta(a \cup a + a \cup_1 \delta a + b) \\
&= \int_{\partial M_4} \delta v \cup (a \cup a + a \cup_1 \delta a + b),
\end{aligned}
\tag{33}
$$

the boundary Hamiltonian contains

$$
-\left( \prod_{e \supset v} X_e \right) (-1)^{\int_{\partial M_4} \delta v \cup (a \cup a + a \cup_1 \delta a + b)}.
\tag{34}
$$

Similarly, for $b \to b + \delta e$ with a boundary edge $e$, we have

$$
\int_{M_4} \phi(a, b + \delta e) - \phi(a, b) = \int_{M_4} \delta e \cup_1 \delta b + b \cup \delta e + \delta e \cup b = \int_{\partial M_4} \delta e \cup_1 b,
\tag{35}
$$

and a boundary term

$$
-\left( \prod_{f \supset e} X_f \right) (-1)^{\int_{\partial M_4} \delta e \cup_1 x}.
\tag{36}
$$

The boundary parts $\phi_3$ and $\phi_3'$ in Eq. (30) and boundary terms Eq. (34), (36) will be described as fermionic particles and fermionic strings after gauging the one-form and two-form symmetries, described in Section 3.3.

## 3.2 Beyond group cohomology invertible phase by gauging symmetry

Next, we gauge the one-form and two-form $\mathbb{Z}_2$ symmetries. We introduce extra $\mathbb{Z}_2$ degrees of freedom on each face $f$ and tetrahedron $\tau$. They can be acted by Pauli matrices $X^A, X^B$ and similar for $Y, Z$. Denote $A = (1 - Z_f^A)/2$ and $B = (1 - Z_t^B)/2$. The Hamiltonian for the gauged model can be constructed by the following steps.

First, we add a term that correlates the gauge transformations for $a, b$ and $A, B$ at low energy

$$
-\sum_e X_e \prod_{f \supset e} X_f^A - \sum_f X_f \prod_{t \supset f} X_t^B,
\tag{37}
$$

where the first product is over all faces sharing the common edge $e$, and the second product is over all tetrahedral sharing the common face $f$. The above term with large coefficient imposes the Gauss law

$$\text{low energy subspace}: \quad X_e \prod X_f^A = 1, \quad X_f \prod X_t^B = 1. \tag{38}$$

Then, we minimally couple the original Hamiltonian to $A, B$ such that the system is invariant under the combined gauge transformation *i.e.* commute with the Gauss law constraint

$$a \to a + \lambda_1, \quad A \to A + \delta\lambda_1, \quad b \to b + \lambda_2, \quad B \to B + \delta\lambda_2, \tag{39}$$

for general $\mathbb{Z}_2$ 1-cochain $\lambda_1$ and 2-cochain $\lambda_2$. This amounts to replacing $\delta b \to \delta b + B$, $\delta a \to \delta a + A$. We also add the flux terms to the Hamiltonian

$$-\sum_t \prod_{f \supset t} Z_f^A - \sum_{\text{4-simplex } p} \prod_{t \subset p} Z_t^B. \tag{40}$$

They impose the flat condition energetically

$$\text{low energy subspace}: \quad \prod_{f \subset t} Z_f^A = 1, \quad \prod_{t \subset p} Z_t^B = 1. \tag{41}$$

They are equivalent to the flat condition $\delta A = 0, \delta B = 0$. Mathematically, the condition $\delta A = 0, \delta B = 0$ is required for gauge fields $A, B$ to obey the cocycle conditions on overlaps of multiple coordinate patches.[10]

To sum up, the bulk Hamiltonian for the beyond group cohomology SPT phase $w_2 w_3$ is

$$
\begin{aligned}
H_{\text{gauged}} = &-\sum_f X_f \prod_t (Z_t^B W_t)^{\int t \cup_2 \delta f} \prod_{f'} (Z_{f'}^A W_{f'})^{\int f' \cup f} \\
&-\sum_e X_e \prod_f (Z_f^A W_f)^{\int f \cup (e \cup_1 \delta e)} \prod_t (Z_t^B W_t)^{\int e \cup t} \\
&\times \prod_{f_1, f_2} CZ(Z_{f_1}^A W_{f_1}, Z_{f_2}^A W_{f_2})^{\int e \cup (f_1 \cup_1 f_2) + f_1 \cup (f_2 \cup_2 \delta e)} \\
&-\sum_e X_e \prod_{f \supset e} X_f^A - \sum_f X_f \prod_{t \supset t} X_t^B - \sum_t \prod_{f \subset t} Z_f^A - \sum_p \prod_{t \subset p} Z_t^B.
\end{aligned}
\tag{43}
$$

We can enforce the Gauss law constraint Eq. (38) strictly, and use the gauge transformation to fix $a = 0, b = 0$ ($Z_e = Z_f = 1$), which implies $W_f = \prod Z_e = 1$ and $W_t = \prod Z_f = 1$, and replace

$$X_e \to \prod_{f \supset e} X_f^A, \quad X_f \to \prod_{t \supset f} X_t^B. \tag{44}$$

Then we arrive at the effective Hamiltonian

$$
\begin{aligned}
H_{w_2 w_3} = &-\sum_f \prod_{t \supset f} X_t^B \prod_{t'} Z_{t'}^{B \int t' \cup_2 \delta f} \prod_{f'} Z_{f'}^{A \int f' \cup f} \\
&-\sum_e \prod_{f \supset e} X_f^A \prod_f Z_f^{A \int f \cup (e \cup_1 \delta e)} \prod_t Z_t^{B \int e \cup t} \prod_{f_1, f_2} CZ(Z_{f_1}^A, Z_{f_2}^A)^{\int e \cup (f_1 \cup_1 f_2) + f_1 \cup (f_2 \cup_2 \delta e)} \\
&-\sum_t \prod_{f \subset t} Z_f^A - \sum_p \prod_{t \subset p} Z_t^B.
\end{aligned}
$$

---

[10] To enforce the zero flux condition, we can also conjugate each Hamiltonian term by a local projector onto the zero flux subspace in the vicinity of the term. That is, for a Hamiltonian term whose support[11] is contained in the bounded region $R$, we conjugate by a projector:

$$\mathcal{P}_R^{\text{0-flux}} \equiv \prod_{t \in R} \frac{(1 + W_t)}{2}, \tag{42}$$

where the product is over tetrahedra in $R$. In the examples we discussed, the flux term will commute with the Hamiltonian, and thus this is not necessary.

Table 1: In the process of gauging the one-form symmetry in 4d, the generators of local, 1-form symmetric operators are mapped according to the duality above. The symmetry operators $A_\Sigma$, the product of $X_e$ on all edges intersecting with a closed codimension-1 surface $\Sigma$, are mapped to the identity in the dual theory. The system on the right-hand side has a $\mathbb{Z}_2$ two-form symmetry, generated by membrane operators $M_\sigma$, where $\sigma$ is a closed 2d surface on the direct lattice.

| Model with $\mathbb{Z}_2$ one-form symmetry | Model with dual $\mathbb{Z}_2$ two-form symmetry |
|:---:|:---:|
| $X_e$ | $\displaystyle\prod_{f \supset e} X_f^A$ |
| $W_f = \displaystyle\prod_{e \subset f} Z_e$ | $Z_f^A$ |
| $A_\Sigma = \displaystyle\prod_{e \perp \Sigma} X_e,\ \delta\Sigma = 0$ | $1$ |
| $1$ | $M_\sigma = \displaystyle\prod_{f \subset \sigma} Z_f^A,\ \partial\sigma = 0$ |

The gauging procedure as an operational replacement is summarized in Table 1 and Table 2.

Now, we give the ground state wavefunction for the $w_2 w_3$ phase. The operational replacement in Table 1 and Table 2 corresponds to the following transformation on states:

$$|a, b\rangle \rightarrow |\delta a, \delta b\rangle, \tag{45}$$

where the entries $\delta a$ and $\delta b$. From the one-form and two-form SPT wavefunction Eq. (26), the $w_2 w_3$ wavefunction is simply

$$\left|\Psi_{w_2 w_3}\right\rangle = \sum_{a,b} e^{i\pi \int \phi_4(a,b)} |\delta a, \delta b\rangle. \tag{46}$$

Thus the wavefunction is a sum of closed surfaces and closed loops weighted by suitable minus signs. One can verify the ground state is unique.[12]

### 3.3 Boundary state

As discussed in Section 3.2, the theory on a closed space after gauging the one-form and two-form symmetries has the bulk Hamiltonian Eq. (45):

$$
\begin{aligned}
H_{w_2 w_3} = &-\sum_f \prod_{t \supset f} X_t \prod_{t'} Z_{t'}^{\int t' \cup_2 \delta f} \prod_{f'} Z_{f'}^{\int f' \cup f} \\
&-\sum_e \prod_{f \supset e} X_f \prod_{f'} Z_{f'}^{\int f' \cup (e \cup_1 \delta e)} \prod_t Z_t^{\int e \cup t} \prod_{f_1, f_2} CZ(Z_{f_1}, Z_{f_2})^{\int e \cup (f_1 \cup_1 f_2) + f_1 \cup (f_2 \cup_2 \delta e)} \\
&-\sum_t \prod_{f \subset t} Z_f - \sum_p \prod_{t \subset p} Z_t,
\end{aligned}
\tag{47}
$$

where we have dropped the superscripts $A, B$ for simplicity. For the boundary Hamiltonian, one strategy is to study the amplitudes of wavefunction, such as Eq. (30), (34), (36). In

---

[12]One way to see this is that $\phi_5(A, B)$ fixes an optimal configuration of $A, B$ with minimal energy and any other configuration of $A, B$ will violate the equation of motion and therefore has an energy cost.

Table 2: In the process of gauging the one-form symmetry in 4d, the generators of local, one-form symmetric operators are mapped according to the duality above. The symmetry operators $A_\Sigma$, the product of $X_e$ on all edges intersecting with a closed codimension-2 surface $\Sigma$, are mapped to the identity in the dual theory. The system on the right-hand side has a $\mathbb{Z}_2$ three-form symmetry, generated by membrane operators $M_\sigma$, where $\sigma$ is a closed 3d surface on the direct lattice.

| Model with $\mathbb{Z}_2$ two-form symmetry | Model with dual $\mathbb{Z}_2$ three-form symmetry |
|---|---|
| $X_f$ | $\prod_{t \supset f} X_t^B$ |
| $W_t = \prod_{f \subset t} Z_f$ | $Z_t^B$ |
| $A_\Sigma = \prod_{f \perp \Sigma} X_f,\ \delta\Sigma = 0$ | $1$ |
| $1$ | $M_\sigma = \prod_{t \subset \sigma} Z_t^B,\ \partial\sigma = 0$ |

this section, we will instead focus on the Hamiltonian level, which is more straightforward to visualize the gauging procedure near the boundary. At the end, we show that these two approaches give the same boundary terms.

We are going to consider the case with boundary $N_3 = \partial M_4 \neq 0$. Our strategy follows the auxiliary vertex approach in Ref. [42], which is reviewed below. To apply our bulk formalism to the case with boundary, we construct a closed manifold by introducing auxiliary simplices: these simplices are given by first introducing a vertex $v_0$, and then introducing auxiliary simplices by connecting $v_0$ to all vertices on the boundary $N_3$ of $M_4$, forming auxiliary edges $\langle 0i \rangle$ $\forall i \in N_3$, auxiliary faces $\langle 0ij \rangle$ $\forall \langle ij \rangle \in N_3$, etc... The space formed by these auxiliary simplices and $N_3$ is denoted as $CN_3$, i.e., the cone over topological space $N_3$. Gluing $CN_3$ and $M_4$ gives a closed spatial 4-manifold:

$$\widetilde{M}_4 \equiv M_4 \sqcup CN_3, \quad \partial\widetilde{M}_4 = 0. \tag{48}$$

We can apply our formalism on this new manifold. For every $a \in C^1(M_4, \mathbb{Z}_2)$, we can extend it to $C^1(\widetilde{M}_4, \mathbb{Z}_2)$ by simply taking $a(e) = 0$ for auxiliary edges $e$. Similarly, we extend $b \in C^2(M_4, \mathbb{Z}_2)$ to $C^2(\widetilde{M}_4, \mathbb{Z}_2)$ by setting $b(f) = 0$ for auxiliary faces $f$. After gauging the one-form and two-form symmetries, the state $|a|_{\widetilde{M}_4}, b|_{\widetilde{M}_4}\rangle$ is mapped to:

$$|\delta a|_{\widetilde{M}_4}, \delta b|_{\widetilde{M}_4}\rangle = |\delta a|_{M_4}, \delta b|_{M_4}, a|_{N_3}, b|_{N_3}\rangle, \tag{49}$$

where $(\cdots)|_S$ means the cochain is restricted to space $S$ and we project $\delta a(\langle 0ij \rangle)$ on the auxiliary face $\langle 0ij \rangle$ to the edge degree of freedom $a(\langle ij \rangle)$ on $N_3$ (since we have taken $a(\langle 0i \rangle) = 0$). The similar story holds for $\delta b|_{\widetilde{M}_4}$ and $b|_{N_3}$. We define the Pauli matrices acting on four entries of (49) as $Z_f, Z_t, \mathcal{Z}_e, \mathcal{Z}_f$.

Now, we are going to study the terms in (47) near $N_3$:

1. $f = \langle 0ij \rangle$ is the auxiliary face: let $e = \langle ij \rangle$ be the corresponding boundary edge.

$$\prod_{t \supset f} \widetilde{X}_t \prod_t \widetilde{Z}_t^{\int_{\widetilde{M}_4} t \cup_2 \delta f} \prod_{f'} \widetilde{Z}_{f'}^{\int_{\widetilde{M}_4} f' \cup f} = \prod_{f|_{f \supset e}^{f \in N_3},} \mathcal{X}_f \prod_{f' \in N_3} \mathcal{Z}_{f'}^{\int_{N_3} \delta e \cup_1 f'}, \tag{50}$$

where we have used

$$
\begin{aligned}
t \cup_2 \delta f(01234) &= t(0123)\delta f(1234) + t(0134)\delta f(1234) \\
&\quad + t(0123)\delta f(0134) + t(0234)\delta f(0124) \\
&= f'(123)\delta e(134) + f'(234)\delta e(124) = \delta e \cup_1 f'(1234).
\end{aligned}
\tag{51}
$$

2. $e = \langle 0i \rangle$ is the auxiliary edge: let $v = \langle i \rangle$ be the corresponding boundary vertex.

$$
\begin{aligned}
&\prod_{f \supset e} \widetilde{X}_f \prod_{f'} \widetilde{Z}_{f'}^{\int_{\widetilde{M}_4} f' \cup (e \cup_1 \delta e)} \prod_t \widetilde{Z}_t^{\int_{\widetilde{M}_4} e \cup t} \prod_{f_1, f_2} CZ(\widetilde{Z}_{f_1}, \widetilde{Z}_{f_2})^{\int_{\widetilde{M}_4} e \cup (f_1 \cup_1 f_2) + f_1 \cup (f_2 \cup_2 \delta e)} \\
&= \prod_{e | \substack{e \in N_3, \\ e \supset v}} \mathcal{X}_e \prod_{t \in N_3} Z_t^{\int_{N_3} v \cup t} \prod_{f_1, f_2 \in N_3} CZ(Z_{f_1}, Z_{f_2})^{\int_{N_3} v \cup (f_1 \cup_1 f_2)} \\
&= \prod_{e | \substack{e \in N_3, \\ e \supset v}} \mathcal{X}_e \prod_{f \in N_3} \mathcal{Z}_f^{\int_{N_3} v \cup \delta f} \prod_{e_1, e_2 \in N_3} CZ(\mathcal{Z}_{e_1}, \mathcal{Z}_{e_2})^{\int_{N_3} v \cup (\delta e_1 \cup_1 \delta e_2)} \\
&= \prod_{e | \substack{e \in N_3, \\ e \supset v}} \mathcal{X}_e \prod_{f \in N_3} \mathcal{Z}_f^{\int_{N_3} \delta v \cup f} \prod_{e_1, e_2 \in N_3} CZ(\mathcal{Z}_{e_1}, \mathcal{Z}_{e_2})^{\int_{N_3} \delta v \cup (e_1 \cup e_2 + e_1 \cup_1 \delta e_2)}.
\end{aligned}
\tag{52}
$$

3. $f = \langle ijk \rangle$ is the boundary face:

$$
\begin{aligned}
&\prod_{t \supset f} \widetilde{X}_t \prod_t \widetilde{Z}_t^{\int_{\widetilde{M}_4} t \cup_2 \delta f} \prod_{f'} \widetilde{Z}_{f'}^{\int_{\widetilde{M}_4} f' \cup f} \\
&= \left( \mathcal{X}_f \prod_{t | \substack{t \in M_4, \\ t \supset f}} X_t \right) \left( \prod_{f' \in N_3} \mathcal{Z}_{f'}^{\int_{N_3} f \cup_1 f' + f' \cup_2 \delta f} \prod_{t \in M_4} Z_t^{\int_{M_4} t \cup_2 \delta f} \right) \\
&\quad \times \left( \prod_{e' \in N_3} \mathcal{Z}_{e'}^{\int_{N_3} e' \cup f} \prod_{f' \in M_4} Z_{f'}^{\int_{M_4} f' \cup f} \right),
\end{aligned}
\tag{53}
$$

where we have used

$$
\begin{aligned}
t \cup_2 \delta f(01234) &= t(0123)\delta f(1234) + t(0134)\delta f(1234) \\
&\quad + t(0123)\delta f(0134) + t(0234)\delta f(0124) \\
&= (f'(123) + f'(134))\delta f(1234) \\
&\quad + f'(123)f(134) + f'(234)f(124) \\
&= (f' \cup_2 \delta f + f \cup_1 f')(1234).
\end{aligned}
\tag{54}
$$

Using $f' \cup_2 \delta f + f \cup_1 f' = \delta(f' \cup_2 f) + \delta f' \cup_2 f + f' \cup_1 f$, we have

$$
\prod_{f' \in N_3} \mathcal{Z}_{f'}^{\int_{N_3} f \cup_1 f' + f' \cup_2 \delta f} = \prod_{f' \in N_3} \mathcal{Z}_{f'}^{\int_{N_3} f' \cup_1 f + \delta f' \cup_2 f}.
\tag{55}
$$

The boundary term (53) becomes

$$
\mathcal{X}_f \prod_{f' \in N_3} \mathcal{Z}_{f'}^{\int_{N_3} f' \cup_1 f + \delta f' \cup_2 f} \prod_{e' \in N_3} \mathcal{Z}_{e'}^{\int_{N_3} e' \cup f} \times \text{bulk terms},
\tag{56}
$$

where the bulk terms only involve $X_t, Z_t$ and $X_f, Z_f$ in $M_4$.

4. $e = \langle ij \rangle$ is the boundary edge:

$$\prod_{f \supset e} \widetilde{X}_f \prod_{f'} \widetilde{Z}_{f'}^{\int_{\widetilde{M}_4} f' \cup (e \cup_1 \delta e)} \prod_t \widetilde{Z}_t^{\int_{\widetilde{M}_4} e \cup t} \prod_{f_1, f_2} CZ(\widetilde{Z}_{f_1}, \widetilde{Z}_{f_2})^{\int_{\widetilde{M}_4} e \cup (f_1 \cup_1 f_2) + f_1 \cup (f_2 \cup_2 \delta e)} \tag{57}$$

$$= \left( \mathcal{X}_e \prod_{f | \substack{f \in M_4, \\ f \supset e}} X_f \right) \left( \prod_{e' \in N_3} \mathcal{Z}_{e'}^{\int_{N_3} e' \cup (e \cup_1 \delta e)} \prod_{f \in M_4} Z_f^{\int_{M_4} f \cup (e \cup_1 \delta e)} \right) \left( \prod_{t \in M_4} Z_t^{\int_{M_4} e \cup t} \right)$$

$$\times \left( \prod_{e_1, e_2 \in N_3} CZ(\mathcal{Z}_{e_1}, \mathcal{Z}_{e_2})^{\int_{N_3} e_1 \cup (\delta e_2 \cup_2 \delta e)} \prod_{f_1, f_2 \in M_4} CZ(Z_{f_1}, Z_{f_2})^{\int_{M_4} e \cup (f_1 \cup_1 f_2) + f_1 \cup (f_2 \cup_2 \delta e)} \right)$$

$$= \mathcal{X}_e \prod_{e' \in N_3} \mathcal{Z}_{e'}^{\int_{N_3} e' \cup (e \cup_1 \delta e)} \prod_{e_1, e_2 \in N_3} CZ(\mathcal{Z}_{e_1}, \mathcal{Z}_{e_2})^{\int_{N_3} e_1 \cup (\delta e_2 \cup_2 \delta e)} \times \text{ bulk terms}, \tag{58}$$

where we have used $e(0 \cdots) = 0$ and

$$\begin{aligned}
f_1 \cup (f_2 \cup_2 \delta e)(01234) &= f_1(012) f_2(234) \delta e(234) \\
&= e_1(12) f_2(234) \delta e(234) \\
&= e_1 \cup (f_2 \cup_2 \delta e)(1234).
\end{aligned} \tag{59}$$

We observe that the bulk term on the boundary gives the open version of the fermionic loop creation operator $U_e^M$.

To summarize, the boundary Hamiltonian is

$$\begin{aligned}
H_{\text{boundary}} = &- \sum_{e \in N_3} \prod_{f | \substack{f \in N_3, \\ f \supset e}} \mathcal{X}_f \prod_{f' \in N_3} \mathcal{Z}_{f'}^{\int_{N_3} \delta e \cup_1 f'} \\
&- \sum_{v \in N_3} \prod_{e | \substack{e \in N_3, \\ e \supset v}} \mathcal{X}_e \prod_{f \in N_3} \mathcal{Z}_f^{\int_{N_3} \delta v \cup f} \prod_{e_1, e_2 \in N_3} CZ(\mathcal{Z}_{e_1}, \mathcal{Z}_{e_2})^{\int_{N_3} \delta v \cup (e_1 \cup e_2 + e_1 \cup_1 \delta e_2)} \\
&- \sum_{f \in N_3} \left( \prod_{e \supset f} \mathcal{Z}_e \right) Z_f - \sum_{t \in N_3} \left( \prod_{f \supset t} \mathcal{Z}_f \right) Z_t \\
&- \sum_{f \in N_3} \mathcal{X}_f \prod_{f' \in N_3} \mathcal{Z}_{f'}^{\int_{N_3} f' \cup_1 f + \delta f' \cup_2 f} \prod_{e' \in N_3} \mathcal{Z}_{e'}^{\int_{N_3} e' \cup f} \times \text{ bulk terms} \\
&- \sum_{e \in N_3} \mathcal{X}_e \prod_{e' \in N_3} \mathcal{Z}_{e'}^{\int_{N_3} e' \cup (e \cup_1 \delta e)} \prod_{e_1, e_2 \in N_3} CZ(\mathcal{Z}_{e_1}, \mathcal{Z}_{e_2})^{\int_{N_3} e_1 \cup (\delta e_2 \cup_2 \delta e)} \times \text{ bulk terms},
\end{aligned} \tag{60}$$

where we only show the part involving $\mathcal{X}_e, \mathcal{Z}_e$ and $\mathcal{X}_f, \mathcal{Z}_f$ on 3d space boundary $N_3 = \partial M_4$. Notice that the first and the second lines above correspond to Eq. (36) and Eq. (34) after gauging the one-form and two-form symmetries. The $\prod_{e \supset v} X_e$ and $\prod_{f \supset e} X_f$ parts in Eqs. (34), (36) contain $X_e, X_f$ in the bulk (overlapping with the boundary partially). After gauging the symmetries, the bulk terms cancel out and the remaining terms completely live on the boundary, as shown in Eq. (60). The last two lines correspond to $\phi_3$ and $\phi_3'$ in Eq. (30):

$$\begin{aligned}
\phi_3 &= b \cup_1 f + \delta b \cup_2 f + a \cup f, \\
\phi_3' &= a \cup [(\delta a \cup_2 \delta e) + e \cup_1 \delta e],
\end{aligned} \tag{61}$$

where we have identified $\mathcal{Z}_e = (-1)^{a(e)}$ and $\mathcal{Z}_f = (-1)^{b(f)}$.

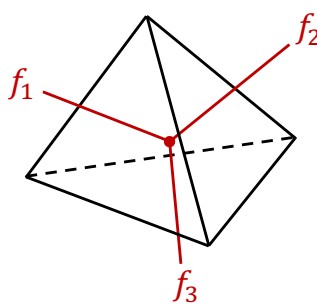

Figure 4: It can be checked that for any set of faces $f_1$, $f_2$, $f_3$ on a tetrahedron, there is always a minus sign in $U_{f_1} U_{f_2} U_{f_3} = -U_{f_3} U_{f_2} U_{f_1}$ [42]. Therefore, we conclude that $U_f$ is the hopping operator of a fermionic particle.

When we drop the bulk terms, the boundary Hamiltonian is not a sum of commuting terms. By defining

$$\mathcal{A}_v = \prod_{e|{e\in N_3, \atop e\supset v}} \mathcal{X}_e \prod_{f\in N_3} \mathcal{Z}_f^{\int_{N_3} \delta v\cup f} \prod_{e_1,e_2\in N_3} CZ(\mathcal{Z}_{e_1}, \mathcal{Z}_{e_2})^{\int_{N_3} \delta v\cup(e_1\cup e_2 + e_1\cup_1 \delta e_2)}, \quad \mathcal{B}_f = \left(\prod_{e\supset f} \mathcal{Z}_e\right) Z_f \,,$$

$$\mathcal{A}_e = \prod_{f|{f\in N_3, \atop f\supset e}} \mathcal{X}_f \prod_{f'\in N_3} \mathcal{Z}_{f'}^{\int_{N_3} \delta e\cup_1 f'}, \quad \mathcal{B}_t = \left(\prod_{f\supset t} \mathcal{Z}_f\right) Z_t \,,$$

the commuting projector Hamiltonian on the boundary $N_3$ is

$$H_\partial = -\sum_v \mathcal{A}_v - \sum_f \mathcal{B}_f - \sum_e \mathcal{A}_e - \sum_t \mathcal{B}_t \,, \tag{62}$$

and we treat the last two lines of Eq. (60) as excitation operators:

$$U_f \equiv \mathcal{X}_f \prod_{f'\in N_3} \mathcal{Z}_{f'}^{\int_{N_3} f'\cup_1 f} \prod_{e'\in N_3} \mathcal{Z}_{e'}^{\int_{N_3} e'\cup f} \,,$$

$$U_e \equiv \mathcal{X}_e \prod_{e'\in N_3} \mathcal{Z}_{e'}^{\int_{N_3} e'\cup(e\cup_1 \delta e)} \prod_{e_1,e_2\in N_3} CZ(\mathcal{Z}_{e_1}, \mathcal{Z}_{e_2})^{\int_{N_3} e_1\cup(\delta e_2\cup_2 \delta e)} \,, \tag{63}$$

where we have dropped $\mathcal{Z}_{f'}^{\int_{N_3} \delta f'\cup f}$ in the definition of $U_f$ since we $\prod_{f'\in t} \mathcal{Z}_{f'} = Z_t$ can be consider as the bulk.

The operator $U_f$ on a face $f$ anti-commutes with two $\mathcal{B}_t$ operators on its adjacent tetrahedra $t$, creating two point-like excitations. Thus, we interpret this $U_f$ as the hopping term of a particle. On the other hand, The operator $U_e$ on an edge $e$ anti-commutes $\mathcal{B}_f$ on faces around the edge $e$, which creates a loop excitation surrounding $e$.

In the following, we will discuss the statistics of the particle and loop excitations on the (3+1)D boundary. We will first show that they have $\pi$ mutual statistics, and the particle is a fermion. In the next section 3.4, we will argue that the loop excitation is also fermionic in the sense of Ref. [30, 33, 48].

**Mutual $\pi$ statistics**    The operators $U_e, U_f$ satisfy

$$U_e U_f = U_f U_e (-1)^{\int_{N_3} e\cup f} \,. \tag{64}$$

Thus the particle excitation created by $U_f$ and the loop excitation created by $U_e$ have mutual $\pi$ statistics.

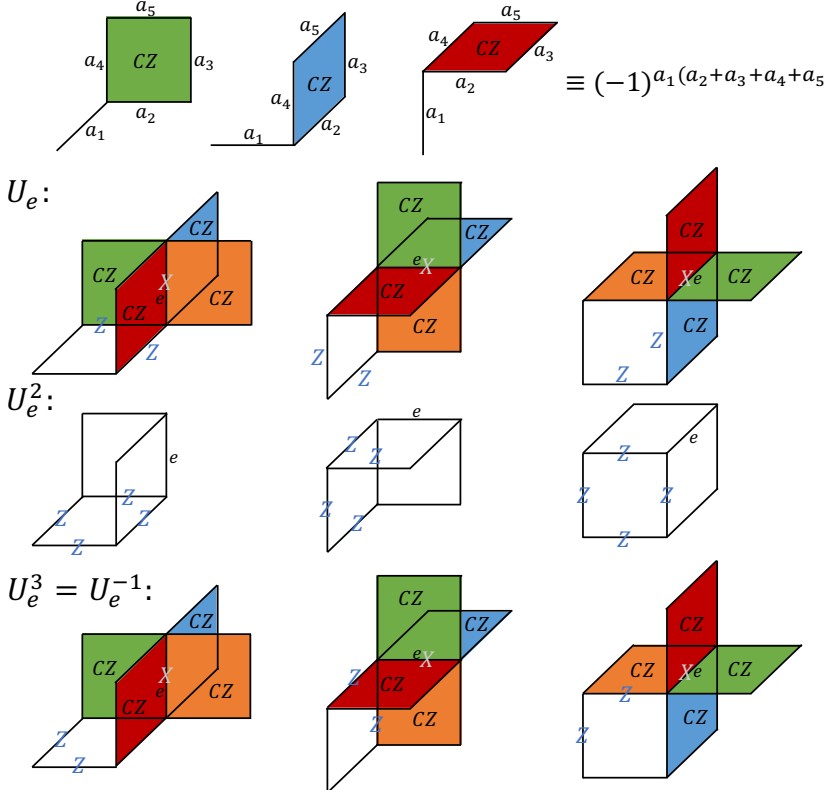

Figure 5: The operator $U_e$ on the cubic lattice.

**fermionic particle $\pi$ self-statistics**    We can detect the fermionic statistic of $U_f$ operator by calculating its commutation relation [49,50]:

$$U_{f_1} U_{f_2} = (-1)^{\int_{N_3} f_1 \cup_1 f_2 + f_2 \cup_1 f_1} U_{f_2} U_{f_1}. \tag{65}$$

In Ref. [49], this operator is shown to have the statistic as the fermionic hopping operator $S_f \equiv i \gamma_{L(f)} \gamma'_{R(f)}$ ($L(f)$ and $R(f)$ are two tetrahedra adjacent to $f$), where the Majorana fermions live at the centers of tetrahedra. Another way to show fermionic statistic is to compute the $T$-junction process [51] directly. Let $f_1, f_2, f_3$ be faces on a tetrahedron $t$. Using Eq. (65), we can check

$$U_{f_1} U_{f_2} U_{f_3} = -U_{f_3} U_{f_2} U_{f_1}, \tag{66}$$

shown in Fig. 4. This minus sign is independent of the choice for branching structures on the tetrahedron [42]. We conclude that the particle excitation is fermionic.

**Fermion loop self-statistics**    In section 3.4 we will discuss a T-junction process for the "fermionic" loop excitation created by $U_e$.

   We remark that an alternative approach to obtain the boundary state is starting from the boundary state of the Hamiltonian model (29) for the "parent" group cohomology phase, and then gauging the one-form and two-form symmetry on the boundary and in the bulk. We leave the detail in Appendix B.

### 3.4   Statistics of fermionic loop excitations on the boundary

In this section, we will discuss a T shape process on the lattice that can describe whether a loop excitation is fermionic. We will use the following property of the fermionic loop excitation.

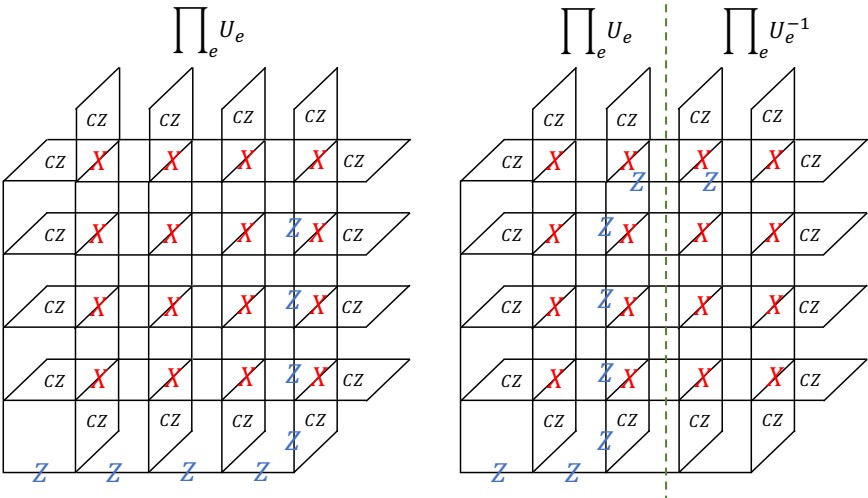

Figure 6: Interface on the membrane operator $\prod U_e$ with the orientation reversed on one side of the interface. We use the ordering convention in the product of operators such that the operators that are diagonal in the $Z$ eigenbasis (the $Z$ and $CZ$ operators in the figure) appear on the right of the $X$ operators, and the product takes the form $\prod X \prod Z \prod CZ$.

The worldsheet of the fermionic loop excitation depends on the volume $\mathcal{V}$ that bounds the worldsheet by $\int_{\mathcal{V}} w_3(TM) = \int_{\mathcal{V}} \frac{dw_2(TM)}{2}$, where we need to take a lift of $w_2(TM)$ from $\mathbb{Z}_2$ to $\mathbb{Z}_4$. In the discussion, we will take the lift to be the value $0, 1$ in $\mathbb{Z}_4$. We note that if we reverse the orientation of (the tangent bundle of) $\mathcal{V}$,

$$\int_{\mathcal{V}} \frac{dw_2(TM)}{2} \to -\int_{\mathcal{V}} \frac{dw_2(TM)}{2} = \int_{\mathcal{V}} \frac{dw_2(TM)}{2} - \int_{\partial \mathcal{V}} w_2(TM). \tag{67}$$

The last term describes a fermionic particle: the world line of a fermionic particle depends on the surface $\Sigma$ that bounds the worldline by $\int_{\Sigma} w_2(TM)$. Thus when we reverse the orientation on the worldsheet of the loop excitation, the fermionic loop has an additional fermionic particle. We can then use the statistics of the fermionic particle, which can be described by a $T$-junction process as in Ref. [51], to describe the statistics of the fermionic loop.[13]

First, the explicit form of $U_e$ on the cubic lattice is shown as Fig. 5, which follows directly from the definition of higher cup products on the cubic lattice [26, 49]. The product of $U_e$ on edges perpendicular to a plane is shown in Figs. 6, 7. We can see that in the bulk of the plane, there is only a product $X_e$ on perpendicular edges, which is the same as the magnetic membrane operator in the (3+1)D toric code. However, the operator $U_e$ here has order 4 ($U_e^4 = 1$), and therefore we can consider the domain wall between opposite orientations $\prod_e U_e$ and $\prod_e U_e^{-1}$, shown in Figs. 6, 7. In this circumstance, there is an additional $Z$ string along the domain wall in the bulk of the plane. This additional $Z$ string represents that this is actually the fermionic hopping operator on the (2+1)D domwain wall [52]

$$\text{On 2d domain wall } D: \quad S_e = X_e Z_{r(e)}, \quad S_e S_{e'} = S_{e'} S_e (-1)^{\int_D e \cup e' + e' \cup e}, \tag{68}$$

---

[13]A related process that describes fermionic loops is proposed in Ref. [27] using Klein bottle, which also involves orientation reversal.



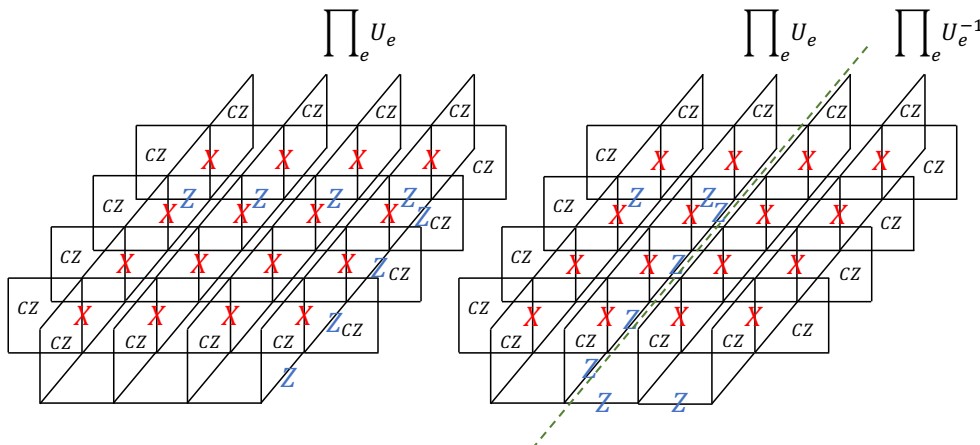

Figure 7: The interface on the membrane operator $\prod U_e$ with the orientation reversed on one side of the interface. We use the ordering convention in the product of operators such that the operators that are diagonal in the $Z$ eigenbasis (the $Z$ and $CZ$ operators in the figure) appear on the right of the $X$ operators, and the product takes the form $\prod X \prod Z \prod CZ$.

where $r(e)$ means the edge starting from the end of $e$ and pointing in the right direction, and 2d means two spatial dimensions. The T-junction process [51] in Fig. 8 also shows the particle is a fermion. To simplify the discussion, we take the planes to be semi-infinite, and all other parts in the figure commute.

## 3.5 Lattice Model for phase transitions

Starting from the Hamiltonian that describes the invertible phase, we can construct a lattice model which is not exactly solvable, but in various limits it can be solved and give different phases. Let us start with the Hamiltonian that described the beyond group cohomology invertible phase

$$
\begin{aligned}
H_{\text{Invertible}} = &-\sum_f \prod_{t \supset f} X_t \prod_t Z_t^{\int t \cup_2 \delta f} \prod_{f'} Z_{f'}^{\int f' \cup f} \\
&-\sum_e \prod_{f \supset e} X_f \prod_f Z_f^{f \cup (e \cup_1 \delta e)} \prod_{f_1, f_2} CZ(Z_{f_1}, Z_{f_2})^{\int e \cup (f_1 \cup_1 f_2) + f_1 \cup (f_2 \cup_2 \delta e)} \prod_t Z_t^{\int e \cup \mathbf{t}} \quad (69) \\
&-\sum_t \prod_{f \subset t} Z_f - \sum_p \prod_{t \subset p} Z_t \, .
\end{aligned}
$$

We add transverse field terms that do not commute with the above Hamiltonian for the invertible phase

$$
H = H_{\text{Invertible}} - \kappa_1 \sum_t Z_t - \kappa_2 \sum_f Z_f \, . \quad (70)
$$

The phase diagram is depicted in Figure 9:

- In the limit $\kappa_1 \to \infty$, while $\kappa_2 = 0$, $Z_t = 1$ on all tetrahedral, and Hamiltonian is left with terms in the second and third lines that contain operators acting on faces. In the Euclidean action, setting the three-form gauge field to zero gives a two-form $\mathbb{Z}_2$ gauge

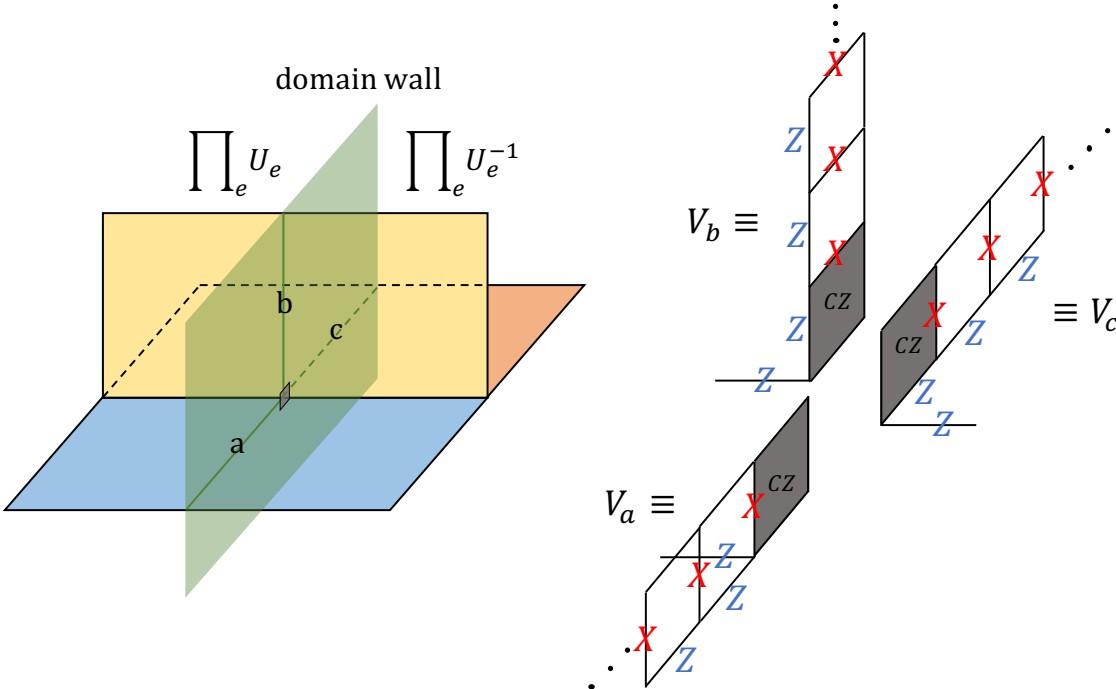

Figure 8: T junction process for the fermion hopping operator on the domain wall as in Ref. [51], where we compare $V_a V_c V_b$ with $V_b V_c V_a$ and find a minus sign. This shows the loop excitation is fermionic. We use the ordering convention in the product of operators such that the operators that are diagonal in the $Z$ eigenbasis (the $Z$ and $CZ$ operators in the figure) appear on the right of the $X$ operators, and the product takes the form $\prod X \prod Z \prod CZ$.

theory in (4+1)D with fermionic magnetic loop, fermionic dyonic loop, and bosonic electric loop excitations.

- In the limit $\kappa_2 \to \infty$, while $\kappa_1 = 0$, $Z_f = 1$ on all faces, and the Hamiltonian is left with terms in the first and third lines that contain operators acting on tetrahedral. In the Euclidean action, setting the two-form gauge field to zero gives a three-form $\mathbb{Z}_2$ gauge theory in (4+1)D with fermionic particle, which is also equivalent to $\mathbb{Z}_2$ one-form gauge theory, but the gauge field is "dynamical spin structure" that is summed over in the path integral.

- In the limit $\kappa_1, \kappa_2 \to \infty$, while all other coefficients are tuned to zero, all degrees of freedoms are frozen and we are left with trivial bulk phase.

Thus there must be one or multiple phase transitions at finite $\kappa_1, \kappa_2$. In particular, the phase transition separating $\kappa_1, \kappa_2 = 0$ and $\kappa_1, \kappa_2 \to \infty$ is protected by the non-trivial invertible phase with effective action $\pi \int w_2 \cup w_3$. This is the analogue of free fermion critical point in (2+1)D protected by the jump in the thermal Hall response when varying the mass from positive to negative. Here, $\kappa_1, \kappa_2$ play the analogue role of the fermion mass parameter. The robustness of the transition does not rely on the presence of symmetries. This represents new phase transitions in (4+1)D. We will revisit such transitions in the future.

The phase transitions across $\kappa_1 = 0$ and $\kappa_1 \to \infty$ at fixed $\kappa_2 = 0$, and the phase transitions across $\kappa_2 = 0$ and $\kappa_2 \to \infty$ at fixed $\kappa_1 = 0$, are ordered-disordered transitions associated with non-invertible and invertible topological orders. There are also phase transitions along

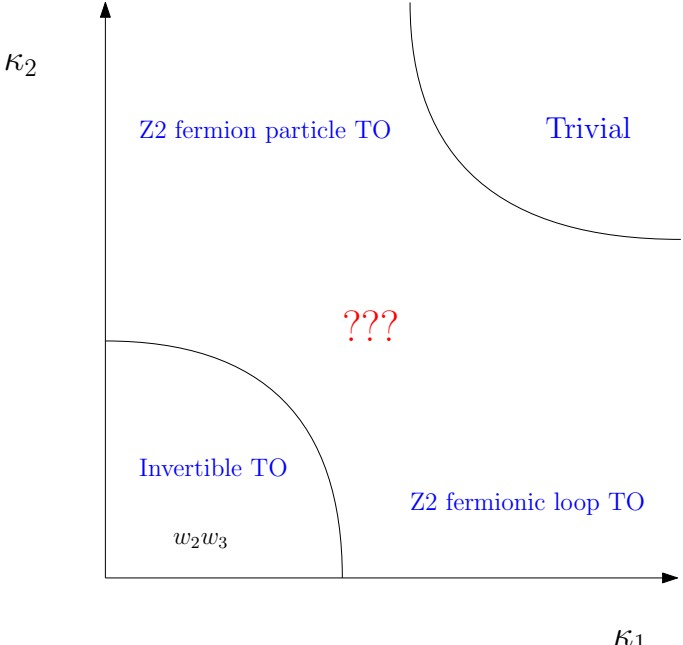

Figure 9: Sketch of the phase diagram of the Hamiltonian model (70) in (4+1)D. The phase transition(s) along the 45 degree line $\kappa_1 = \kappa_2$ are protected by the non-trivial invertible topological order on the lower left corner. The $\mathbb{Z}_2$ fermionic loop topological order in (4+1)D has three non-trivial basic excitations: electric loop, magnetic loop, and dyonic loop, where the magnetic and dyonic loop excitations are fermionic while the electric loop excitation is bosonic (similar theory with only bosonic loop excitations is discussed in Ref. [53]).

$\kappa_1 = -\kappa_2$ that are associated with changing the topological order from fermionic particles to fermionic loop excitations.

We note that the invertible phase has boundary $\mathbb{Z}_2$ topological order with fermionic particle and fermionic loop excitations in (3+1)D, and they have mutual statistics. The boundary of (4+1)D $\mathbb{Z}_2$ fermionic particle topological order can also have fermionic particle on the gapped boundary where the magnetic membrane excitation condenses. The $\mathbb{Z}_2$ fermionic loop topological order in (4+1)D also has gapped boundary with $\mathbb{Z}_2$ fermionic loop, where the magnetic loop excitation condenses.

## 3.6 Application: non-trivial quantum cellular automaton in (4+1)D

A quantum cellular automaton (QCA) is an automorphism on the algebra of local operators acting on the Hilbert space of an infinite quantum system. It is still an open question to classify non-trivial QCAs above 3+1 dimensions.[14] It has many applications to topological phases of matter, see *e.g.* Refs. [12, 47, 54–56].[15]

In this section, we explicitly construct a new quantum cellular automaton from the bulk $w_2 w_3$ Hamiltonian (45) in the (4+1)D hypercubic lattice. We will study this $w_2 w_3$ Hamiltonian and show that it can be transformed to locally flippable separators, which is equivalent to a QCA [47].

---

[14]A non-trivial QCAs means that it is not simply a translational operation or conjugation by a finite-depth local unitary quantum circuit.

[15]For instance, Ref. [47] uses non-trivial QCA to disentangle the ground state of invertible phases with non-trivial boundary state.

The non-triviality of the QCA can be argued as follows. We have shown that the boundary theory of this $w_2 w_3$ Hamiltonian contains fermionic particle and fermionic loop excitations with $\pi$ mutual statistic. Based on the framing argument at the end of Section 1, this theory can not be consistently defined in strictly (3+1)D lattices. This corresponds to the anomalous (3+1)D $\mathbb{Z}_2$ topological orders in Ref. [33]. Following the same reasoning in Ref. [47], suppose this QCA were trivial, *i.e.* a finite-depth local unitary quantum circuit, we would have a commuting projector Hamiltonian for the anomalous (3+1)D topological order, which is contradicted with the previous arguments. Therefore, this (4+1)D QCA from the $w_2 w_3$ Hamiltonian should be a non-trivial QCA.

We note that the above argument is based on the assumption that the boundaries for the bulk invertible phase with a boundary gravitational anomaly, such as the boundary $\mathbb{Z}_2$ gauge theory in (3+1)D with fermionic particles and fermionic strings with $\pi$ mutual statistics, cannot be realized by a local commuting projector Hamiltonian that can be defined in purely (3+1)D fashion.

First, to simplify the bulk Hamiltonian Eq. (45), we define the shorthand notation (to simplify the notations, we have omitted the superscripts $A, B$ on the Pauli operators that act on the degrees of freedom living on faces and tetrahedrons, respectively)

$$G_f \equiv \prod_{t \supset f} X_t \prod_{t'} Z_{t'}^{\int t' \cup_2 \delta f} . \tag{71}$$

Then, we rewrite the Hamiltonian Eq. (45) on hypercubic lattice as the following Hamiltonian that has the same ground state:[16]

$$H'_{w_2 w_3} = -\sum_{f'} Z_{f'} \prod_f G_f^{\int f' \cup f} \tag{72}$$

$$-\sum_t Z_t \prod_e \left[ \prod_{f \supset e} X_f \prod_{f'} Z_{f'}^{\int f' \cup (e \cup_1 \delta e)} \prod_{f_1, f_2} CZ(Z_{f_1}, Z_{f_2})^{\int e \cup (f_1 \cup_1 f_2) + f_1 \cup (f_2 \cup_2 \delta e)} \right]^{\int e \cup t},$$

where we have modified the stabilizers to contain single $Z_f$ or $Z_t$. Next, we substitute $Z_f$ as the product of $G_{f'}$ in the second term

$$H'_{w_2 w_3} = -\sum_{f'} Z_{f'} \prod_f G_f^{\int f' \cup f} - \sum_t Z_t \prod_e \left[ \prod_{f \supset e} X_f \prod_{f'} (\prod_{f''} G_{f''}^{\int f' \cup f''})^{\int f' \cup (e \cup_1 \delta e)} \right.$$

$$\left. \times \prod_{f_1, f_2} CZ(\prod_{f'_1} G_{f'_1}^{\int f_1 \cup f'_1}, \prod_{f'_1} G_{f'_2}^{\int f_2 \cup f'_2})^{\int e \cup (f_1 \cup_1 f_2) + f_1 \cup (f_2 \cup_2 \delta e)} \right]^{\int e \cup t} \tag{73}$$

$$\equiv -\sum_f \widehat{Z}_f - \sum_t \widehat{Z}_t .$$

For this Hamiltonian, we can easily find the flippers as

$$\widehat{X}_f \equiv X_f , \qquad \widehat{X}_t \equiv X_t \prod_{t'} Z_{t'}^{\int t \cup_2 t'} . \tag{74}$$

---

[16]On the hypercubic lattice, the Hamiltonian $H'_{w_2 w_3}$ can be obtained by rearranging the first two terms in Eq. (45) using the property that the pairing of two faces $f, f'$ that have non-trivial cup product $\int f \cup f' \neq 0$ gives a one-to-one correspondence between $f$ and $f'$. In particular, the $Z$ flux terms are dropped, since they are generated by the first two terms of $H'_{w_2 w_3}$.

Notice that $\widehat{X}_t$ commute with all $G_{f'}$, so the commutation relations are simply

$$
\begin{aligned}
\widehat{X}_f \widehat{Z}_{f'} &= (-1)^{\delta_{f,f'}} \widehat{Z}_{f'} \widehat{X}_f, \qquad \widehat{X}_t \widehat{Z}_{t'} = (-1)^{\delta_{t,t'}} \widehat{Z}_{t'} \widehat{X}_t, \\
[\widehat{X}_f, \widehat{X}_t] &= [\widehat{X}_f, \widehat{Z}_t] = [\widehat{Z}_f, \widehat{X}_t] = [\widehat{Z}_f, \widehat{Z}_t] = 0.
\end{aligned}
\tag{75}
$$

We remark that although the discussion in this subsection is on the hypercubic lattice, since the $H'_{w_2 w_3}$ can also be defined on arbitrary triangulated lattice, we expect the model also gives a non-trivial QCA on any triangulated lattice of spatial dimension four.

# 4 Beyond group cohomology SPT phase with $\mathbb{Z}_2$ symmetry in (4+1)D

In this section we will apply our construction to the beyond group cohomology bosonic SPT phase with $\mathbb{Z}_2$ unitary symmetry in (4+1)D. It has the following effective action

$$
\pi \int \mathcal{C}_1 \cup w_2^2 = \pi \int (\mathcal{C}_1 \cup w_2) \cup w_2,
\tag{76}
$$

where $\mathcal{C}_1$ is a $\mathbb{Z}_2$-valued one-form background field for the $\mathbb{Z}_2$ symmetry. Note other effective actions of the background fields belong to the "cohomology SPT phases", since they can be expressed in terms of the background $\mathcal{C}_1$ of the $\mathbb{Z}_2$ symmetry.[17]

A lattice Hamiltonian model for such beyond group cohomology SPT phase has been constructed in Ref. [12]. It uses the property that the domain wall generating the $\mathbb{Z}_2$ symmetry is decorated with $\pi \int w_2^2 = -\frac{1}{24\pi} \int \mathrm{Tr}\, R \wedge R$ mod $2\pi$, which can be written as a gravitational Chern-Simons term on the boundary, and thus the boundary of the domain wall has thermal Hall conductance characterized by chiral central charge $c = 4$ mod 8. An example with such chiral central charge is the three-fermion theory, and the domain wall in the model of Ref. [12] is described by the three-fermion Walker Wang model. However, in general there is no simple description of chiral central charge from the lattice Hamiltonian (there are expressions for the change of chiral central charge), and the construction in Ref. [12] cannot be generalized to other beyond group cohomology SPT phases in an obvious way. Thus we will revisit the problem using the general construction of the lattice Hamiltonian described in Section 2.

## 4.1 "Parent" group cohomology SPT phase

The SPT phase can be constructed similar to Section 3. Here we start with the SPT phase with $\mathbb{Z}_2$ 0-form, one-form, and two-form symmetries with the effective action $\pi \int \phi_5$ with

$$
\phi_5(\mathcal{C}_1, A_2, B_3) = B_3 \cup_1 B_3 + A_2 \cup B_3 + \mathcal{C}_1 \cup A_2 \cup A_2.
\tag{77}
$$

Integrating out $B_3$ imposes $A_2 = w_2$ (since $B_3 \cup_1 B_3 = w_3 \cup B_3$), and thus we recover the effective action $\pi \int \mathcal{C}_1 \cup w_2 \cup w_2$. We note that $\phi_5$ satisfies

$$
\phi_5(\delta c, \delta a, \delta b) = \delta \phi_4(c, a, b), \quad \phi_4(c, a, b) = b \cup b + b \cup_1 \delta b + a \cup \delta b + c \cup \delta a \cup \delta a.
\tag{78}
$$

We introduce $\mathbb{Z}_2$ degrees of freedom on each vertex, edge and face. They are acted by the Pauli matrices $X_v, X_e, X_f$, and similarly for $Y, Z$, respectively. The Hamiltonian for the SPT phase is then

$$
\begin{aligned}
H_{\text{parent}} = &-\sum_v X_v (-1)^{\int \phi_4(c+v,a,b) - \phi_4(c,a,b)} - \sum_e X_e (-1)^{\int \phi_4(c,a+e,b) - \phi_4(c,a,b)} \\
&- \sum_f X_f (-1)^{\int \phi_4(c,a,b+f) - \phi_4(c,a,b)}.
\end{aligned}
\tag{79}
$$

---

[17]Explicitly, $w_3 \cup \mathcal{C}_1^2 = 0$ due to $\mathcal{C}_1^2 = Sq^1 \mathcal{C}_1, w_3 = Sq^1 w_2$, and $\int \mathcal{C}_1^3 \cup w_2 = \int Sq^2(\mathcal{C}_1^3) = \int \mathcal{C}_1 \cup (\mathcal{C}_1^2)^2 = \int \mathcal{C}_1^5$.

Explicitly,

$$
\begin{aligned}
\phi_4(c+v,a,b) - \phi_4(c,a,b) &= v \cup \delta a \cup \delta a\,, \\
\phi_4(c,a+e,b) - \phi_4(c,a,b) &= e \cup \delta b + \delta c \cup (\delta a \cup_1 \delta e) + \delta[c \cup (\delta a \cup_1 \delta e)]\,, \\
\phi_4(c,a,b+f) - \phi_4(c,a,b) &= \delta b \cup_2 \delta f + \delta a \cup f + \delta \phi_3\,, \\
\phi_3 &= b \cup_1 f + \delta b \cup_2 f + a \cup f\,.
\end{aligned}
\tag{80}
$$

We note that the phase factor in the Hamiltonian depends on $c, a, b$ only through $\delta c, \delta a, \delta b$. This will be important when we gauge the symmetries.

## 4.2 Beyond group cohomology SPT phase by gauging symmetry

Let us proceed to gauge the one-form and two-form symmetries. We introduce new $\mathbb{Z}_2$ degrees of freedom on each face and tetrahedron. Denote the Pauli matrices by $X_f^A, X_t^B$ and similarly for $Y, Z$. Following the procedure in Section 3.2, the effective Hamiltonian after gauging the symmetry is

$$
\begin{aligned}
H_{\mathcal{C}w_2^2} = &-\sum_v X_v \prod_{f_1,f_2} CZ(Z_{f_1}^A, Z_{f_2}^A)^{\int v \cup f_1 \cup f_2} - \sum_e \prod_{f \supset e} X_f^A \prod_t Z_t^{B \int e \cup t} \prod_{v,f'} CZ(Z_v, Z_{f'}^A)^{\int \delta v \cup (f' \cup_1 \delta e)} \\
&-\sum_f \prod_{t \supset f} X_t^B \prod_t Z_t^{B \int t \cup_2 \delta f} \prod_{f'} Z_{f'}^{A \int f' \cup f} \\
&-\sum_t \prod_{f \subset t} Z_f^A - \sum_{4\text{-simplex } p} \prod_{t \subset p} Z_t^B\,.
\end{aligned}
\tag{81}
$$

The Hamiltonian is a sum of local commuting projectors. The 0-form symmetry is $\prod_v X_v$. We will drop the superscripts $A, B$ for simplicity.

## 4.3 Boundary states

### 4.3.1 Anomalous $\mathbb{Z}_2$ topological order with $\mathbb{Z}_2$ symmetry

We will discuss an anomalous $\mathbb{Z}_2$ topological order in (3+1)D with $\mathbb{Z}_2$ 0-form symmetry, which can be a boundary state for the beyond group cohomology invertible phase.

Consider $\mathbb{Z}_2$ gauge theory in (3+1)D with the action

$$
\pi \int a \cup \delta b + \pi \int a \cup w_2 \cup \mathcal{C}_1 + \pi \int w_2 \cup b\,,
\tag{82}
$$

where $a$ is $\mathbb{Z}_2$ 1-cochain, $b$ is $\mathbb{Z}_2$ 2-cochain. Equivalently, we turn on background $w_2 \cup A$ for the $\mathbb{Z}_2$ two-form symmetry that acts on $\oint b$, and background $w_2$ for the $\mathbb{Z}_2$ one-form symmetry that acts on $\oint a$. The equation of motion of $a$ gives $\delta b = w_2 \cup \mathcal{C}_1$. This implies that under a background gauge transformation of the $\mathbb{Z}_2$ symmetry,

$$
\mathcal{C}_1 \to \mathcal{C}_1 + \delta \lambda\,, \quad b \to b + w_2 \cup \lambda\,,
\tag{83}
$$

where $\lambda = 0, 1$ is $\mathbb{Z}_2$ 0-cochain. In particular, if we perform non-trivial transformation $\lambda = 1$ on half of $\int b_2$, where $b_2$ on this half is shifted by extra $w_2$, then the interface in the middle that separates the regions with $\lambda = 0$ and with $\lambda = 1$ lives on the boundary of $\int w_2$. Thus the interface describes the worldline of a fermionic particle bound to $\int b_2$.

### 4.3.2 New boundary theory from lattice Hamiltonian model

We use the same boundary construction as Section 3.3. The original manifold $M_4$ has boundary $N_3 = \partial M_4$. We introducing an auxiliary vertex $v_0$, connecting to all vertices on the boundary. The union of $M_4$ and this cone of $N_3$ is a closed manifold $\widetilde{M}_4 \equiv M_4 \sqcup CN_3$, and we can define the $\mathcal{C}w_2^2$ Hamiltonian on this closed manifold. We can further project faces and tetrahedra in $CN_3$ to edges and faces in $N_3$, such as Eq. (49). Therefore, the total degrees of freedom are $Z_v, Z_f, Z_t \quad \forall v, f, t \in M_4$, and $\mathcal{Z}_e, \mathcal{Z}_f \quad \forall e, f \in N_3$, with additional $\mathcal{Z}_{v_0}$ at the auxiliary vertex (the global symmetry is $\mathcal{X}_{v_0} \prod_v X_v = 1$ and $\mathcal{Z}_{v_0}$ can be fixed to $+1$ by applying the symmetry action, so we can get rid of the dependence on $v_0$). We now study the terms in (81) near $N_3$:

1. $v = \langle 0 \rangle$ is the auxiliary vertex:

$$-\mathcal{X}_{v_0} \prod_{f_1, f_2} CZ(\widetilde{Z}_{f_1}, \widetilde{Z}_{f_2})^{\int_{\widetilde{M}_4} v_0 \cup f_1 \cup f_2} = -\prod_{v \in M_4} X_v \prod_{e_1, e_2 \in N_3} CZ(\mathcal{Z}_{e_1}, \mathcal{Z}_{e_2})^{\int_{N_3} e_1 \cup \delta e_2}. \tag{84}$$

   Notice that we have used the property $\mathcal{X}_{v_0} = \prod_{v \in M_4} X_v$ by the global symmetry constraint.

2. $e = \langle 0i \rangle$ is the auxiliary edge: let $v = \langle i \rangle$ be the corresponding boundary vertex.

$$\prod_{f \supset e} \widetilde{X}_f \prod_t \widetilde{Z}_t^{\int_{\widetilde{M}_4} e \cup t} \prod_{v, f'} CZ(\widetilde{Z}_v, \widetilde{Z}_{f'})^{\int_{\widetilde{M}_4} \delta v \cup (f' \cup_1 \delta e)} = \prod_{e | e \in N_3, \atop e \supset v} \mathcal{X}_e \prod_{t \in N_3} Z_t^{\int_{N_3} v \cup t}. \tag{85}$$

3. $f = \langle 0ij \rangle$ is the auxiliary face: let $e = \langle ij \rangle$ be the corresponding boundary edge.

$$\prod_{t \supset f} \widetilde{X}_t \prod_t \widetilde{Z}_t^{\int_{\widetilde{M}_4} t \cup_2 \delta f} \prod_{f'} \widetilde{Z}_{f'}^{\int_{\widetilde{M}_4} f' \cup f} = \prod_{f | f \in N_3, \atop f \supset e} \mathcal{X}_f \prod_{f' \in N_3} \mathcal{Z}_{f'}^{\int_{N_3} \delta e \cup_1 f'}, \tag{86}$$

   which is the same as Eq. (50).

4. $v = \langle i \rangle$ is the boundary vertex:

$$-\widetilde{X}_v \prod_{f_1, f_2} CZ(\widetilde{Z}_{f_1}, \widetilde{Z}_{f_2})^{\int_{\widetilde{M}_4} v \cup f_1 \cup f_2} = -X_v \prod_{f_1, f_2} CZ(Z_{f_1}, Z_{f_2})^{\int_{M_4} v \cup f_1 \cup f_2}$$
$$= -X_v \times \text{ bulk terms}. \tag{87}$$

5. $e = \langle ij \rangle$ is the boundary edge:

$$\prod_{f \supset e} \widetilde{X}_f \prod_t \widetilde{Z}_t^{\int_{\widetilde{M}_4} e \cup t} \prod_{v, f'} CZ(\widetilde{Z}_v, \widetilde{Z}_{f'})^{\int_{\widetilde{M}_4} \delta v \cup (f' \cup_1 \delta e)}$$

$$= \left( \mathcal{X}_e \prod_{f | f \in M_4, \atop f \supset e} X_f \right) \left( \prod_{t \in M_4} Z_t^{\int_{M_4} e \cup t} \right)$$

$$\times \left( \prod_{v, e' \in N_3} CZ(Z_v, \mathcal{Z}_{e'})^{\int_{N_3} v \cup (\delta e' \cup_1 \delta e)} \prod_{v, f' \in M_4} CZ(Z_v, Z_{f'})^{\int_{M_4} \delta v \cup (f' \cup_1 \delta e)} \right)$$

$$= \mathcal{X}_e \prod_{v, e' \in N_3} CZ(Z_v, \mathcal{Z}_{e'})^{\int_{N_3} v \cup (\delta e' \cup_1 \delta e)} \times \text{ bulk terms}. \tag{88}$$

6. $f = \langle ijk \rangle$ is the boundary face:

$$
\prod_{t \supset f} \widetilde{X}_t \prod_t \widetilde{Z}_t^{\int_{\widetilde{M}_4} t \cup_2 \delta f} \prod_{f'} \widetilde{Z}_{f'}^{\int_{\widetilde{M}_4} f' \cup f}
$$

$$
= \left( \mathcal{X}_f \prod_{\substack{t \mid t \in M_4, \\ t \supset f}} X_t \right) \left( \prod_{f' \in N_3} \mathcal{Z}_{f'}^{\int_{N_3} f \cup_1 f' + f' \cup_2 \delta f} \prod_{t \in M_4} Z_t^{\int_{M_4} t \cup_2 \delta f} \right)
$$

$$
\times \left( \prod_{e' \in N_3} \mathcal{Z}_{e'}^{\int_{N_3} e' \cup f} \prod_{f' \in M_4} Z_{f'}^{\int_{M_4} f' \cup f} \right) \tag{89}
$$

$$
= \mathcal{X}_f \prod_{f' \in N_3} \mathcal{Z}_{f'}^{\int_{N_3} f' \cup_1 f + \delta f' \cup_2 f} \prod_{e' \in N_3} \mathcal{Z}_{e'}^{\int_{N_3} e' \cup f} \times \text{bulk terms},
$$

which is the same as Eq. (56).

Similar to the discussion in Section 3.3, we interpret Eqs. (84), (85), (86) as the stabilizers in the boundary Hamiltonian, while Eqs. (87), (88), (89) are treated as the operators for boundary excitation. We are going to study these boundary excitations.

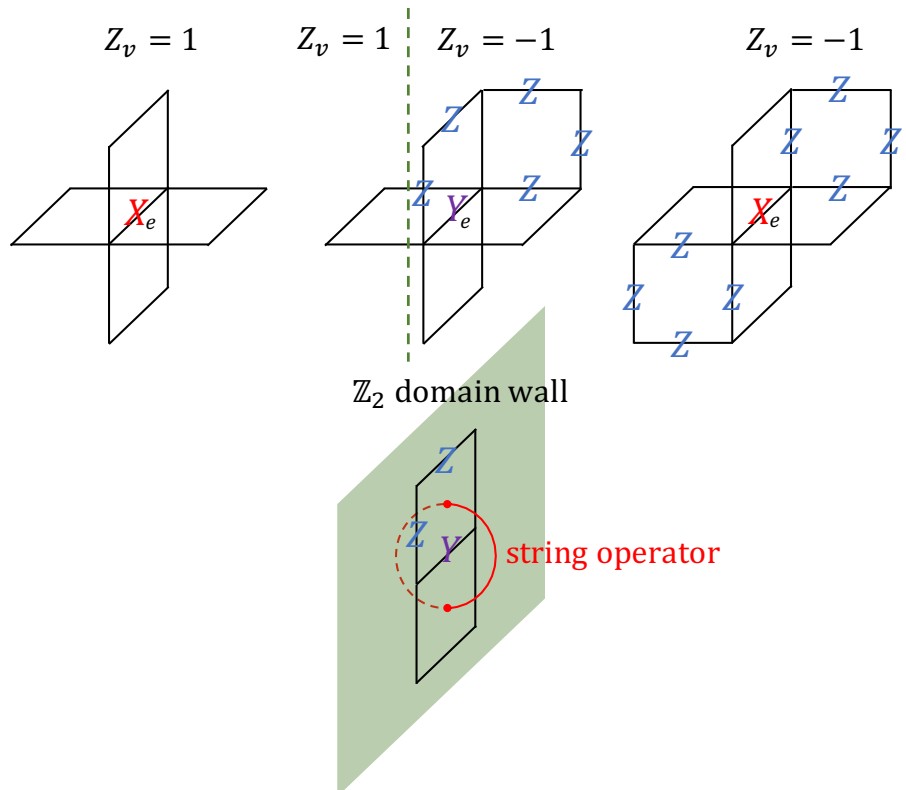

Figure 10: The intersection of the membrane and the domain wall is a fermion hopping operator.

As shown before, Eq. (89) corresponds to the hopping operator of fermionic particles. For the global $\mathbb{Z}_2$ symmetry, we consider it as a background since we didn't gauge this symmetry. Eq. (87) simply means changing the background configuration of $Z_v$. Eq. (88) is the most interesting operator corresponding to a loop excitation. We are going to demonstrate this operator on the cubic lattice and show that intersection points between the oop excitation

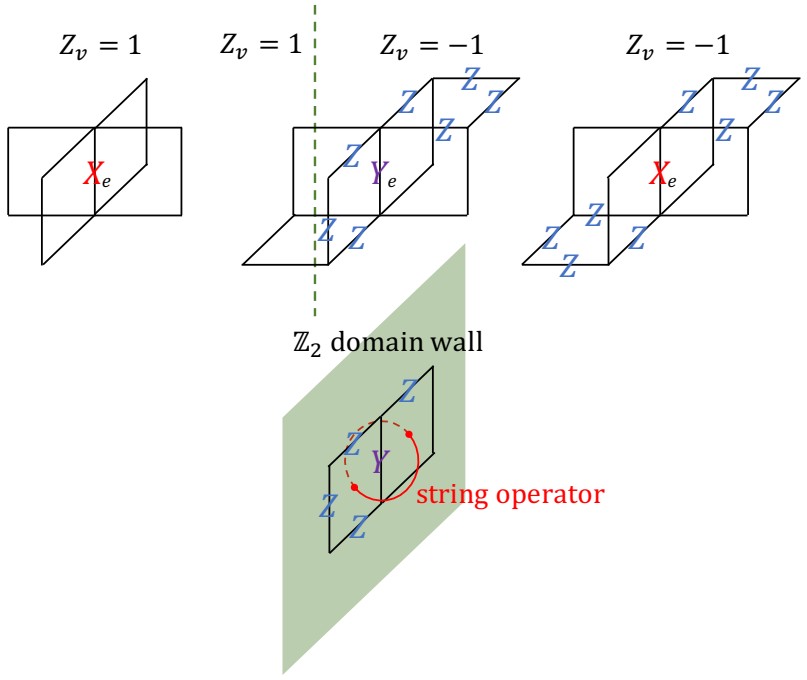

Figure 11: The intersection of the membrane and the domain wall is a fermion hopping operator.

and the $\mathbb{Z}_2$ domain wall have fermionic statistic. Some explicit expressions of the operator $U'_e \equiv X_e \prod_{v,e' \in N_3} CZ(Z_v, Z_{e'})^{\int_{N_3} v \cup (\delta e' \cup_1 \delta e)}$ are shown in Figs. 10, 11. In the region with all $Z_v = 1$, the $CZ$ part in $U'_e$ doesn't contribute and it is simply $U'_e = X_e$. In the region with all $Z_v = -1$, we have $U'_e = X_e \prod_{e'} Z_{e'}^{\int e \cup \delta e' + \delta e' \cup e}$. Notice that this corresponds to the Hamiltonian for $\mathbb{Z}_2$ one-form with the topological action $S = \pi \int B_2 \cup B_2$ [40, 42, 57]. For $U'_e$ on the domain wall between $Z_v = 1$ and $Z_v = -1$, it is drawn in the middle of Figs. 10, 11. We can see this operator creates two excitations on the intersection points between the loop excitation and the $\mathbb{Z}_2$ domain wall. Based on the T-junction process in Fig. 12, this point-like excitation has fermionic statistic.

## 4.4 Comparison with the lattice Hamiltonian model in literature

Let us compare with the lattice Hamiltonian model in Ref. [12]. The lattice models are different, with different degrees of freedom and different Hamiltonian. Ref. [12] constructs the Hamiltonian by decorating the domain wall that generates the $\mathbb{Z}_2$ symmetry with three-fermion Walker Wang model, which has effective action $\pi \int w_2^2$. On the other hand, here we construct the Hamiltonian model using a different invertible gauge theory in (4+1)D.[18] Ref. [12] also discussed a gapped symmetric boundary state by introducing ancilla bosonic degrees of freedom on the boundary that contains a fermion, and condensing the composite boson made out of the extra fermion and a fermion in the three-fermion topological order on the 3d boundary of the domain wall generating the $\mathbb{Z}_2$ symmetry, which is decorated with the three-fermion Walker-Wang model. Here, we directly obtain the related gapped symmetric boundary state by truncating the bulk Hamiltonian.

---

[18]The three-fermion Walker Wang model can also be described by an invertible gauge theory with action $\pi \int (A_2 \cup A_2 + A_2 \cup B_2 + B_2 \cup B_2)$ [21, 26], but the theory differs from what we discuss here, where the background $C_1$ for the $\mathbb{Z}_2$ 0-form symmetry couples to the domain wall $\pi \int A_2 \cup A_2$.

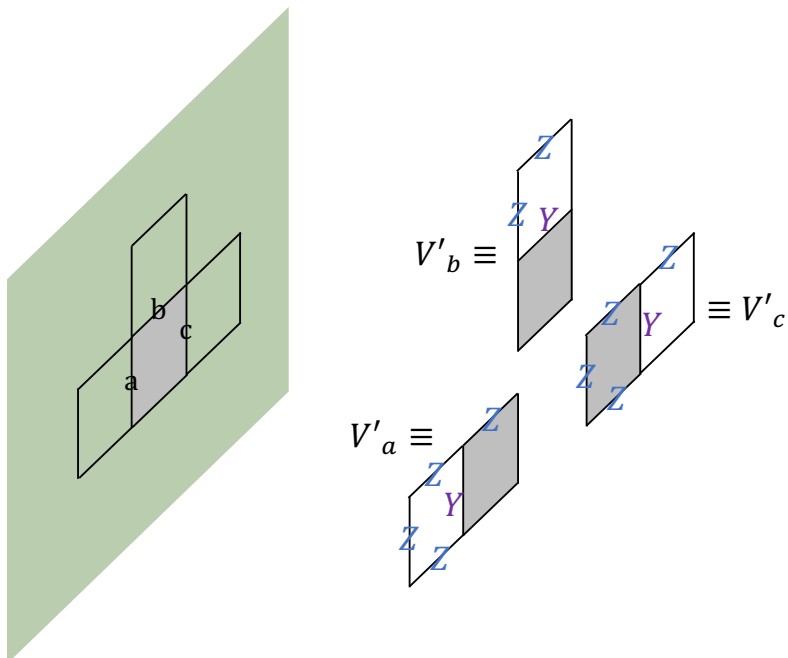

Figure 12: The T junction process for the fermion hoppping operator on the intersection of the loop creation operator and the domain wall: $V'_a V'_c V'_b = -V'_b V'_c V'_a$, which implies that the loop excitation is fermionic.

# 5 Conclusion and future directions

In this note, we describe a general method to construct exactly solvable lattice Hamiltonian model for bosonic beyond group cohomology invertible phases that have order two or order four, in any spacetime dimension. We illustrate the procedure using two examples in (4+1)D: one without symmetry, and the other has $\mathbb{Z}_2$ unitary symmetry.

Let us comment on several generalizations and future questions. Some of them are investigated in future work.

**Short-range entangled phases** The construction generalizes with minor modifications to invertible phases with higher-form symmetry and/or time-reversal symmetry instead of unitary ordinary symmetry.[19]

One future direction is to study the topological invariant associated with the lattice model. Such invariant can be characterized by an anomalous boundary state. In the example of the invertible phase without symmetry in (4+1)D, we can take the invariant to be single pair of fermionic particles and fermionic loop excitation with $\pi$ mutual statistics. Can we compute the numerical topological invariant in terms of the given Hamiltonian, which specifies which invertible phase the Hamiltonian belongs to? This can be similar to the formula for the chiral central charge for gapped Hamiltonian given by Kitaev in Ref. [58].[20]

---

[19]The invertible phases with time-reversal symmetry have effective action that can be probed by non-orientable spacetime whose tangent bundle has non-trivial first Stiefel-Whitney class $w_1(TM)$, which can be represented by the invertible gauge theory

$$\pi \int Sq^1(v_d) + u_1 \cup v_d, \quad Sq^1(v_d) = v_d \cup_{d-1} v_d, \tag{90}$$

with $\mathbb{Z}_2$ 1-cocycle $u_1$ and $d$-cocycle $v_d$ in $(d+1)$ dimensional spacetime. The first term equals $\pi \int w_1(TM) \cup v_d$ by the Wu formula [36], and thus the equation of for $v_d$ implies $u_1 = w_1(TM)$.

[20]See also more recent works [59, 60].

While in this note we describe the beyond group cohomology invertible phases using lattice Hamiltonian model, one can also describe the phases using the ground state wavefunctions as in Ref. [61]. For instance, one can study the invertible phase using the entanglement property of the wavefunction, such as in Ref. [62].

The construction of the lattice Hamiltonian for beyond group cohomology phases can also be applied to construct new "beyond group cohomology" phases with subsystem symmetry.

We can also generalize the lattice models for the beyond group cohomology invertible phases to include foliation structure, and explore the analogue of gravitational anomaly for possible excitations with restricted mobility on the boundary.[21] For instance, is there an analogue of the anomalous $\mathbb{Z}_2$ topological order with fermionic particles and fermionic loop excitations?

Quantum cellular automata (QCA) can characterize non-trivial topological phases of matter [47, 54, 55], such as the Floquet topological phases [55, 56]. In Section 3.6 we argue that our lattice model of invertible phase without symmetry in (4+1)D gives a new non-trivial QCA. It might be interesting to explore the connection between general invertible phases constructed using our method and new non-trivial QCA that can be applied to investigate other phases.

**Long range entangled phases and gapless phases**     We can also couple the invertible gauge theory to models with topological order or gapless models to change the statistics of excitations.[22] This changes not only the bulk but also changes the gapped boundaries, since only excitations with trivial statistics can condense on the boundary [66–72].

One can also explore the lattice process to describe the statistics of the extended excitations such as loops and membranes when they are no longer bosonic excitations with trivial statistics.

The coupling to invertible gauge theory can also change the low energy dynamics; for instance, the statistics of excitations is related to the anomaly of the symmetry generated by the corresponding creation operator [20]. If we change the statistics, the one-form symmetry can become anomalous, and the model cannot be in confining phase with unbroken one-form symmetry.

**Quantum phase transitions**     Another direction is to better understand models with phase transitions obtained by including transverse field terms, in analogue to the toric code in the transverse field studied in Ref. [73]. When the invertible phases do not have symmetry, these are quantum phase transitions robust to any perturbations. Are there continuous phase transitions? Are there deconfined quantum phase transitions? Are there critical points protected by "beyond group cohomology" invertible phases with subsystem symmetry?

## Acknowledgement

We thank Anton Kapustin for discussion and participation at the early stage of the project and a related project. We thank Alexei Kitaev, Ryan Thorngren, Chao-Ming Jian and Ryohei Kobayashi for discussions. We thank Maissam Barkeshli, Xie Chen, Meng Cheng, Tyler Ellison, Anton Kapustin, Alexei Kitaev, Shu-Heng Shao, Wilbur Shirley, Nathanan Tantivasadakarn, and Cenke Xu for comments on a draft. The work of P.-S. H. is supported by the U.S. Department of Energy, Office of Science, Office of High Energy Physics, under Award Number DE-SC0011632, and by the Simons Foundation through the Simons Investigator Award. Y.-A. C is supported by the JQI fellowship at the University of Maryland.

---

[21]This can also be addressed in field theory such as [63, 64].

[22]In (2+1)D quantum field theories, this is discussed in Ref. [65], where the spin of line operator can be changed by activating a background for $\mathbb{Z}_2$ one-form symmetry, with the background set to $w_2(TM)$.

**Note Added**   Near the completion of this work, we learned of another work in preparation [74] that also constructs an exactly solvable lattice Hamiltonian model for the beyond group cohomology invertible phase without symmetry in (4+1)D. In [35] we show that the fermionic loop creation operator can be shown to produces $(-1)$ statistics using the process in [74] that detects loop self statistics.

## A   Review of cochains and cup products

In this appendix, we will review some mathematical properties of cochains and cup products. For more details, see *e.g.* Refs. [36] and [75]. For a review of cochains and higher cup products on the hypercubic lattice $\mathbb{Z}^d$, see Ref. [26,49].

### A.1   Cochain and cup products on triangulation

We triangulate the spacetime manifold $M$ with simplicies, where a $p$-simplex is the $p$-dimensional analogue of a triangle or tetrahedron (for $p = 0$ it is a point, $p = 1$ it is an edge, etc). The $p$-simplices can be described by its vertices $(i_0, i_1, \cdots i_p)$ where we pick an ordering $i_0 < i_1 < \cdots i_p$.

A simplicial $p$-cochain $f \in C^p(G, \mathcal{A})$ is a function on $p$-simplices taking values in an Abelian group $\mathcal{A}$ (we use additive notation for Abelian groups). For simplicity, we will take $\mathcal{A}$ to be a field (an Abelian group endowed with two products: addition and multiplication). In the following we will take $\mathcal{A}$ to be integers.

The coboundary operation on the cochains $\delta : C^p(M, \mathcal{A}) \to C^{p+1}(M, \mathcal{A})$ is defined by

$$(\delta f)(i_0, i_1, \cdots i_{p+1}) = \sum_{j=0}^{p+1} (-1)^j f(i_0, \cdots \widehat{i_j}, \cdots i_{p+1}), \tag{91}$$

where the hatted vertices are omitted. The coboundary operation is nilpotent $\delta^2 = 0$. When a cochain $x$ satisfies $\delta x = 0$, it is called a cocycle.

The cup product $\cup$ for $p$-cochain $f$ and $q$-cochain $g$ gives a $(p+q)$-cochain defined by

$$(f \cup g)(i_0 \cdots i_{p+q}) = f(i_0, \cdots i_p) g(i_p \cdots i_{p+q}). \tag{92}$$

It is associative but not commutative.

The higher cup products can be defined in a similar way [76]. For $p$ cochain $f$ and $q$ cochain $q$, the higher cup product $f \cup_i g$ is a $(p+q-i)$ cochain. The definition for $\cup_1$ is

$$f \cup_1 g(0, 1, \cdots, (p+q-1)) = \sum_{j=0}^{p-1} (-1)^{(p-j)(q+1)} f(0, \cdots j, j+q, \cdots p+q-1) g(j, \cdots j+q). \tag{93}$$

For instance, for an 1-cochain $a$ and 2-cochains $u, v$

$$u \cup_1 v(0123) = u(023)v(012) - u(013)v(123),$$
$$a \cup_1 u(012) = -a(02)u(012), \quad u \cup_1 a(012) = u(012)a(01) + u(012)a(12). \tag{94}$$

### A.2   Cochain and cup product on hypercubic lattice

In this section, we review the definition of higher cup products on the cubic lattice in Refs. [26, 49]. We follow the convention used in Ref. [49]. We assume that all cochains are $\mathbb{Z}_2$-valued.

The cup products on squares □ (faces in Fig. 13) are defined as

$$
\begin{aligned}
\lambda \cup \lambda'(\square_{0123}) &= \lambda(01)\lambda'(13) + \lambda(02)\lambda'(23), \\
\lambda \cup \lambda'(\square_{0145}) &= \lambda(01)\lambda'(15) + \lambda(04)\lambda'(45), \\
\lambda \cup \lambda'(\square_{0246}) &= \lambda(02)\lambda'(26) + \lambda(04)\lambda'(46),
\end{aligned}
\tag{95}
$$

where $\lambda, \lambda \in C^1(M_3, \mathbb{Z}_2)$ are 1-cochains and all other faces are defined using the translational symmetry.

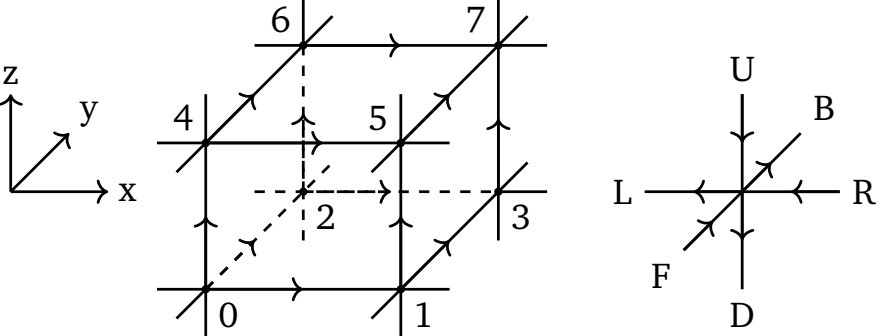

Figure 13: There are six faces for each cube $c$. U,D,F,B,L,R stand for faces on direction "up","down","front","back","left","right". We assign the face U, F, R to be inward and D, B, L to be outward. The $\cup_1$ product on two 2-cochain is defined by $\beta \cup_1 \beta'(c) = \beta(L)\beta'(B) + \beta(L)\beta'(D) + \beta(B)\beta'(D) + \beta(U)\beta'(F) + \beta(U)\beta'(R) + \beta(F)\beta'(R)$.

If $\lambda$ and $\beta$ are a 1-cochain and a 2-cochain, the corresponding cup products are on the cubic lattice are defined as follows:

$$
\begin{aligned}
\lambda \cup \beta(c) &= \lambda(01)\beta(\square_{1357}) + \lambda(02)\beta(\square_{2367}) + \lambda(04)\beta(\square_{4567}), \\
\beta \cup \lambda(c) &= \beta(\square_{0123})\lambda(37) + \beta(\square_{0145})\lambda(57) + \beta(\square_{0246})\lambda(67),
\end{aligned}
\tag{96}
$$

where $c$ is a cube whose vertices are labeled in Fig. 13. For a cup product involving 0-cochains $\alpha$, the definition is straightforward:

$$
\begin{aligned}
\alpha \cup \beta(\square_{0123}) &= \alpha(0)\beta(\square_{0123}), \\
\beta \cup \alpha(\square_{0123}) &= \beta(\square_{0123})\alpha(3), \\
\alpha \cup \lambda(01) &= \alpha(0)\lambda(01), \\
\lambda \cup \alpha(01) &= \lambda(01)\alpha(1),
\end{aligned}
\tag{97}
$$

with other faces defined similarly.

For cup-1 products, we first define between a 1-cochain $\lambda$ and a 2-cochain $\beta$:

$$
\begin{aligned}
\lambda \cup_1 \beta(\square_{0123}) &= [\lambda(02) + \lambda(23)]\beta(\square_{0123}), \\
\beta \cup_1 \lambda(\square_{0123}) &= [\lambda(01) + \lambda(13)]\beta(\square_{0123}),
\end{aligned}
\tag{98}
$$

with the same formula for other two faces by replace numbers $(0,1,2,3)$ with $(0,1,4,5)$ and $(0,2,4,6)$. The cup-1 product between two 2-cochains $\beta, \beta'$ is shown in Fig. 13. The cup-1 product between an 1-cochain $\lambda$ and a 3-cochain $\gamma$ is

$$
\begin{aligned}
\lambda \cup_1 \gamma(c) &= [\lambda(04) + \lambda(46) + \lambda(67)]\gamma(c), \\
\gamma \cup_1 \lambda(c) &= [\lambda(01) + \lambda(13) + \lambda(37)]\gamma(c).
\end{aligned}
\tag{99}
$$

### A.3 Identities for cup products

We summarize some formulas for manipulating cup products. They hold for any lattice geometry. For a $p$-cochain $f$ and $q$-cochain $g$:

$$f \cup_i g = (-1)^{pq-i} g \cup_i f + (-1)^{p+q-i-1} \left( \delta(f \cup_{i+1} g) - \delta f \cup_{i+1} g - (-1)^p f \cup_{i+1} \delta g \right),$$
$$\delta(f \cup_i g) = \delta f \cup_i g + (-1)^p f \cup_i \delta g + (-1)^{p+q-i} f \cup_{i-1} g + (-1)^{pq+p+q} g \cup_{i-1} f. \tag{100}$$

Consider 1-cochain supports on a single edge $e$, denoted by $\boldsymbol{e}$, which takes value 1 on edge $e$ and 0 otherwise. Then from the definition of cup product, we have

$$\boldsymbol{e} \cup F(\boldsymbol{e}, x) = 0 = F(\boldsymbol{e}, x) \cup \boldsymbol{e}, \tag{101}$$

where $F$ is any cochain-valued function that vanishes for $\boldsymbol{e} = 0$ i.e. $F(0, x) = 0$. This generalizes to any cochain $\boldsymbol{e}_n$ that supports on basic simplex of degree $n \geq 1$.

Similarly, from the definition of $\cup_1$ product we have

$$\boldsymbol{f} \cup_1 F(\boldsymbol{f}, x) = 0 = F(\boldsymbol{f}, x) \cup_1 \boldsymbol{f}, \tag{102}$$

where $F$ is any cochain-valued function that vanishes for $\boldsymbol{f} = 0$ i.e. $F(0, x) = 0$. This generalizes to any cochain $\boldsymbol{e}_n$ that supports on basic simplex of degree $n \geq 2$.

## B  Boundary state of invertible gauge theory from gauging symmetries

In this appendix, we obtain the boundary state of invertible gauge theory for the invertible phase without symmetry in (4+1)D, from the boundary state of the "parent" group cohomology SPT phase with $\mathbb{Z}_2$ one-form and two-form symmetries by gauging the symmetries in the bulk and on the boundary.[23]

### B.1  Boundary action

**Boundary action for "parent" group cohomology SPT phase**    The bulk effective action for the "parent" group cohomology SPT phase is equivalent to

$$\pi \int (w_2 \cup B_3 + A_2 \cup B_3 + A_2 \cup w_3), \tag{103}$$

where we have used the Wu formula, and the fields are background fields. Let us derived a gapped boundary action by anomaly inflow. Consider the transformation

$$A_2 \to A_2 + \delta a_1, \quad B_3 \to B_3 + \delta b_2, \quad w_2 \to w_2 + \delta a_1', \quad w_3 \to w_3 + \delta b_2'. \tag{104}$$

The action produces the boundary term

$$\pi \int (a_1 \cup \delta b_2 + w_2 \cup b_2 + a_1 \cup w_3 + a_1 \cup B_3 + A_2 \cup b_2)$$
$$+ \pi \int \left( A_2 \cup b_2' + a_1' \cup B_3 + \delta a_1 \cup b_2' + a_1' \cup \delta b_2 \right). \tag{105}$$

---

[23]A similar method for one-form symmetry SPT phases in 3+1d is discussed in Ref. [20].

The redefinition $a_1' \to a_1' + a_1$ implies that the boundary action describes the product of an ordinary $\mathbb{Z}_2$ gauge theory and $\mathbb{Z}_2$ gauge theory with fermionic particle ("dynamical spin structure")

$$\pi \int \left( \delta a_1 \cup b_2' + A_2 \cup b_2' + a_1 \cup w_3 \right) + \pi \int \left( a_1' \cup \delta b_2 + a_1' \cup B_3 + A_2 \cup b_2 + w_2 \cup b_2 \right), \quad (106)$$

where the first term describes the ordinary $\mathbb{Z}_2$ gauge theory [35]. We note that in the $\mathbb{Z}_2$ gauge theory with fermionic particle, the loop excitations that braid with the fermion are bosonic [35]. In the ordinary $\mathbb{Z}_2$ gauge theory bosonic particle, the fermionic loop excitation [35] braids with boson instead of fermion. Thus the boundary does not have gravitational anomaly.

**Gauging the symmetries**   When we gauge the one-form and two-form symmetries by promoting $A_2, B_3$ to be dynamical fields, the equation of motion for $B_3$ sets $a_1' = 0$, the equation of motion for $A_2$ sets $b_2' = b_2$, and we recover the action for the anomalous $\mathbb{Z}_2$ gauge theory with fermionic particle and fermionic string with mutual braiding:

$$\pi \int \left( \delta a_1 \cup b_2 + a_1 \cup w_3 + w_2 \cup b_2 \right). \quad (107)$$

## B.2   Lattice model

**Boundary Hamiltonian model for "parent" group cohomology SPT phase**   The symmetry generator on the boundary is given by

$$U(v) = \left( \prod_{e \supset v} X_e \right) (-1)^{\int_{M_4} \phi_4(a+\delta v, b) - \phi_4(a,b)}, \quad U'(e) = \left( \prod_{f \supset e} X_f \right) (-1)^{\int_{M_4} \phi_4(a, b+\delta e) - \phi_4(a,b)}, \quad (108)$$

where the products in $U(v), U'(e)$ denote edges met at the vertex $v$ and faces met at the edge $e$, respectively. From Eq.(34), (36), they equal to

$$U(v) = \left( \prod_{e \supset v} X_e \right) (-1)^{\int_{\partial M_4} \delta v \cup (a \cup a + a \cup_1 \delta a + b)}, \quad U'(e) = \left( \prod_{f \supset e} X_f \right) (-1)^{\int_{\partial M_4} \delta e \cup_1 b}, \quad (109)$$

where phase factors only depend on degrees of freedom on the boundary $\partial M_4 = N_3$. If the vertex $v$ and the edge $e$ are in the bulk, the symmetry generators are

$$U(v) = \prod_{e \supset v} X_e, \quad U'(e) = \prod_{f \supset e} X_f. \quad (110)$$

As Ref. [40], we restrict Eq. (109) on the boundary and obtain:

$$U_\partial(v) = \prod_{e|^{e \in N_3,}_{e \supset v}} X_e \prod_f Z_f^{\int_{N_3} \delta v \cup f} \prod_{e_1, e_2} CZ(Z_{e_1}, Z_{e_2})^{\int_{N_3} \delta v \cup (e_1 \cup e_2 + e_1 \cup_1 \delta e_2)},$$

$$U_\partial'(e) = \prod_{f|^{f \in N_3,}_{f \supset e}} X_f \prod_{f'} Z_{f'}^{\int_{N_3} \delta e \cup_1 f'}. \quad (111)$$

A symmetric gapped boundary state is given by the Hamiltonian

$$H_{\text{boundary}} = -\sum_v U_\partial(v) - \sum_e U_\partial'(e) - \sum_f \prod_{e \subset f} Z_e - \sum_t \prod_{f \subset t} Z_f, \quad (112)$$

where we added the last two terms to penalize configurations with nonzero $\delta a, \delta b$. They commute with the first two terms. We will call the first two terms in the Hamiltonian (112) the "electric terms", since the operators that anti-commute with such terms transform under the one-form and two-form symmetries, while the second two terms in the Hamiltonian (112) the "flux terms".[24]

We note that $\int \delta v \cup b_2 = \int v \cup \delta b_2$ and $\int \delta v \cup (a \cup a + a \cup_1 \delta a) = \int v \cup (\delta a \cup_1 \delta a)$ can be expressed in terms of the flux terms, where $a(e) = (1 - Z_e)/2$, $b(f) = (1 - Z_f)/2$, and thus we can rewrite the boundary Hamiltonian as

$$
\begin{aligned}
H'_{\text{boundary}} &= H^{(1)}_{\text{boundary}} + H^{(2)}_{\text{boundary}}, \\
H^{(1)}_{\text{boundary}} &= -\sum_v \prod X_e - \sum_f \prod Z_e, \\
H^{(2)}_{\text{boundary}} &= -\sum_e \prod X_f \prod_{f'} Z_{f'}^{\int_{N_3} \delta e \cup_1 f'} - \sum_t \prod_{f \subset t} Z_f,
\end{aligned}
\tag{113}
$$

where $H^{(1)}_{\text{boundary}}$ is the ordinary $\mathbb{Z}_2$ toric code model and $H^{(2)}_{\text{boundary}}$ is the $\mathbb{Z}_2$ gauge theory with fermionic particle and bosonic string.

In the upcoming work [35] we show that the ordinary 3+1d $\mathbb{Z}_2$ gauge theory with boson particle has bosonic "electric loop", bosonic magnetic flux loop and fermionic "dyonic loop" given by the fusion of the electric and magnetic loops. The non-anomalous $\mathbb{Z}_2$ gauge theory with fermionic particle has fermionic "electric loop", bosonic magnetic loop and bosonic "dyonic loop". In the anomalous $\mathbb{Z}_2$ gauge theory with fermionic particles, all three loops are fermionic. In [35] we show the "electric loop" belongs to the trivial superselection sector, and thus the "dyonic loop" belongs to the same superselection sector as the magnetic loop, but their creation operators have different correlation functions.

**Gauging symmetry** After we gauge the one-form and two-form symmetries, the symmetry generator acts on the Hilbert space as the identity operator, and we look for operators that commute with the symmetry generators $U_\partial(v)$ and $U_\partial(e)$. There are the following fermionic loop creation operator and fermionic particle hopping operator, which have $\pi$ mutual statistics:

$$
\begin{aligned}
U_e^M &= X_e (-1)^{\int_{N_3} a \cup (\delta a \cup_2 \delta e + e \cup_1 \delta e)}, \quad U_f^M = X_f (-1)^{\int_{N_3} a \cup f + b \cup_1 f}, \\
U_e^M U_f^M &= U_f^M U_e^M (-1)^{\int_{N_3} e \cup f}.
\end{aligned}
\tag{114}
$$

Thus we recover the anomalous boundary $\mathbb{Z}_2$ topological order.

## C Boundary state of Toric code model from truncation

To illustrate the truncation of bulk Hamiltonian, let us consider the toric code model in (2+1)D [45]. There are two gapped boundaries [77]: the smooth boundary and the rough boundary, where the magnetic and the electric charge condenses on the boundary, respectively.

The bulk theory has qubit on each edge, with Hamiltonian

$$
H = -\sum_v \prod X_e - \sum_f \prod Z_e.
\tag{115}
$$

---

[24]The construction of symmetry gapped boundary is similar to that in Ref. [40] for one-form symmetry SPT phase in (3+1)D, where the boundary has zero framing anomaly.

We introduce an auxiliary vertex 0, and $Z_{0v} = \widetilde{Z}_v, X_{0v} = \widetilde{X}_v$ for boundary vertex $v$. The boundary terms are

$$\widetilde{X}_v X_{v,v-1} X_{v,v+1} X_{v,b}, \quad \widetilde{Z}_{v-1} \widetilde{Z}_{v+1} Z_{v,v+1}, \quad Z_{v-1,b} Z_{v+1,b} Z_{v,v+1}, \tag{116}$$

where $v-1, v, v+1$ denote consecutive boundary vertices, and $b$ denote bulk vertex. We note $Z_v, X_v$ do not commute with the boundary terms and thus the excitations have energy cost. The construction above corresponds to the rough boundary condition (condensation of $e$ particles). This can be seen from that a $Z_e$ string terminated on the boundary with an addition $\widetilde{Z}_v$ operators on the endpoints commute with all terms in Eq. (116). This operator creates two $e$ particles without energy costs, and thus $e$ condensed on the boundary.

We can consider another boundary construction without introducing new degrees of freedom $\widetilde{X}_v, \widetilde{Z}_v$:

$$X_{v,v-1} X_{v,v+1} X_{v,b}, \quad Z_{v-1,b} Z_{v+1,b} Z_{v,v+1}. \tag{117}$$

This corresponds to the smooth boundary condition (condensation of $m$ particles). The $X_e$ string on the dual lattice terminated on the boundary commutes with all terms in the Hamiltonian. Therefore, two $m$ particles created on the boundary have zero energy costs, and $m$ condenses on the boundary.

# D  Explicit expression of boundary Hamiltonian on the 3d cubic lattice

In this appendix, we explicitly derive the boundary Hamiltonian Eq. (60) on the 3d cubic lattice by truncating the bulk Hamiltonian model for the invertible phase without symmetry in 4+1d. We will show that terms in Eq. (60) can be drawn as Fig. 1 and Fig. 2.

The first term in Eq. (60) is

$$\prod_{f|\substack{f \in N_3, \\ f \supset e}} \mathcal{X}_f \prod_{f' \in N_3} \mathcal{Z}_{f'}^{\int_{N_3} \delta e \cup_1 f'}, \tag{118}$$

which corresponds to the first three terms in Fig. 1, using the definition of higher cup products in Appendix A.2.[25]

The second term in Eq. (60) is

$$\prod_{e|\substack{e \in N_3, \\ e \supset v}} \mathcal{X}_e \prod_{f \in N_3} \mathcal{Z}_f^{\int_{N_3} \delta v \cup f} \prod_{e_1, e_2 \in N_3} CZ(\mathcal{Z}_{e_1}, \mathcal{Z}_{e_2})^{\int_{N_3} \delta v \cup (e_1 \cup e_2 + e_1 \cup_1 \delta e_2)}$$
$$= \prod_{e|\substack{e \in N_3, \\ e \supset v}} \mathcal{X}_e \prod_{f \in N_3} (-1)^{\int_{N_3} \delta v \cup (a \cup a + a \cup_1 \delta a + b)}, \tag{119}$$

where we have labelled $\mathcal{Z}_e = (-1)^{a(e)}$ and $\mathcal{Z}_f = (-1)^{a(f)}$.

Now, we are going to use the following identity:

$$\frac{1}{2}([\delta a]_2 - \delta a) = (a \cup a + a \cup_1 \delta a) \ (\text{mod } 2), \tag{120}$$

---

[25]This term is the same as the gauge constraint in the 3d bosonization discussed in Ref. [49], which enforces the particle excitations in the gauge-invariant low energy subspace to be fermionic.



$$(-1)^{a\cup a + a\cup_1 \delta a}:$$

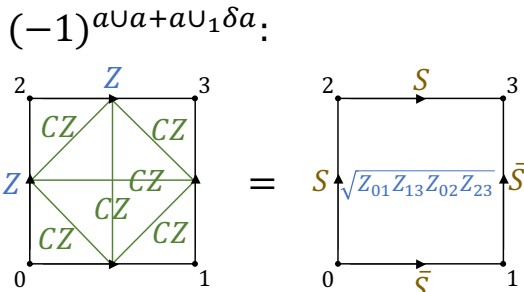

Figure 14: "Electric" string operator.

where $a(e) = 0, 1$ is $\mathbb{Z}_2$-valued, $\delta a(\square_{0123}) \equiv a(01) + a(13) - a(02) - a(23)$ is the coboundary operator on a square $\square_{0123}$ in Fig. 13 (same for $\square_{0145}$ and $\square_{0246}$), and $[\cdots]_2 = 0, 1$ is the mod 2 residue of $\cdots$. By the definition of higher cup products, we have

$$
\begin{aligned}
[a\cup a + a\cup_1 \delta a](\square_{0123}) =& a(02) + a(23) + a(01)a(13) + a(01)a(02) + a(01)a(23) \\
=& + a(13)a(02) + a(13)a(23) + a(02)a(23) \pmod{2},
\end{aligned}
\tag{121}
$$

where $(-1)^{a(e)}$ corresponds to the gate $Z_e$ and $(-1)^{a(e)a(e')}$ corresponds to the controlled-Z gate $CZ(e, e')$, which are shown in the left side of Fig. 14. The identity Eq. (120) can be checked straightforwardly in Table 3. From Eq. (120), we have

Table 3: The table enumerates all cases of $(a_{01}, a_{13}, a_{02}, a_{23})$ and verifies that all of them satisfy Eq. (120).

| $a_{01}$ | $a_{13}$ | $a_{02}$ | $a_{23}$ | $\delta a$ | $[\delta a]_2$ | $a\cup a + a\cup_1 \delta a$ |
|---|---|---|---|---|---|---|
| 0 | 0 | 0 | 0 | 0 | 0 | 0 |
| 0 | 0 | 0 | 1 | -1 | 1 | 1 |
| 0 | 0 | 1 | 0 | -1 | 1 | 1 |
| 0 | 0 | 1 | 1 | -2 | 0 | 1 |
| 0 | 1 | 0 | 0 | 1 | 1 | 0 |
| 0 | 1 | 0 | 1 | 0 | 0 | 0 |
| 0 | 1 | 1 | 0 | 0 | 0 | 0 |
| 0 | 1 | 1 | 1 | -1 | 1 | 1 |
| 1 | 0 | 0 | 0 | 1 | 1 | 0 |
| 1 | 0 | 0 | 1 | 0 | 0 | 0 |
| 1 | 0 | 1 | 0 | 0 | 0 | 0 |
| 1 | 0 | 1 | 1 | -1 | 1 | 1 |
| 1 | 1 | 0 | 0 | 2 | 0 | 1 |
| 1 | 1 | 0 | 1 | 1 | 1 | 0 |
| 1 | 1 | 1 | 0 | 1 | 1 | 0 |
| 1 | 1 | 1 | 1 | 0 | 0 | 0 |

$$
(-1)^{a\cup a + a\cup_1 \delta a(\square_{0123})} = (-1)^{\frac{[\delta a]_2(\square_{0123})}{2} - \frac{\delta a(\square_{0123})}{2}} = \sqrt{Z_{01}Z_{13}Z_{02}Z_{23}} S_{02} S_{23} \bar{S}_{01} \bar{S}_{13},
\tag{122}
$$

where the square root gives $1, i$ and the $S$ gate is

$$
S = \begin{pmatrix} 1 & 0 \\ 0 & i \end{pmatrix},
\tag{123}
$$

and $\overline{S}$ is the complex conjugation of $S$. This operator is shown in the right side of Fig. 14. Therefore, Eq. (119) contains the product of $a \cup a + a \cup_1 \delta a + b$ over 6 faces on a cube and becomes the last diagram in the first row of Fig. 1.

The third line in Eq. (60) contains

$$-\sum_{f \in N_3} \left( \prod_{e \supset f} \mathcal{Z}_e \right) Z_f - \sum_{t \in N_3} \left( \prod_{f \supset t} \mathcal{Z}_f \right) Z_t \,, \tag{124}$$

which become the "flux terms" in the second raw of Fig. 1. We have omitted $Z_f$ and $Z_t$ since we only focus on auxiliary degrees of freedom $\mathcal{X}_e, \mathcal{Z}_e, \mathcal{X}_f, \mathcal{Z}_f$.

The fourth line of Eq. (60) is

$$-\sum_{f \in N_3} \mathcal{X}_f \prod_{f' \in N_3} \mathcal{Z}_{f'}^{\int_{N_3} f' \cup_1 f + \delta f' \cup_2 f} \prod_{e' \in N_3} \mathcal{Z}_{e'}^{\int_{N_3} e' \cup f} \times \text{bulk terms} \,, \tag{125}$$

which can be considered as the hopping term for fermionic particles. We use $\prod_{f' \subset t} \mathcal{Z}_{f'} = Z_t$ and keep only $\mathcal{X}, \mathcal{Z}$ degrees of freedom, the hopping term becomes

$$U_f = \mathcal{X}_f \prod_{f' \in N_3} \mathcal{Z}_{f'}^{\int_{N_3} f' \cup_1 f} \prod_{e' \in N_3} \mathcal{Z}_{e'}^{\int_{N_3} e' \cup f} \,, \tag{126}$$

which is drawn in the first row of Fig. 2 using the definition of higher cup products.

The last term in Eq. (60) is

$$-\sum_{e \in N_3} \mathcal{X}_e \prod_{e' \in N_3} \mathcal{Z}_{e'}^{\int_{N_3} e' \cup (e \cup_1 \delta e)} \prod_{e_1, e_2 \in N_3} CZ(\mathcal{Z}_{e_1}, \mathcal{Z}_{e_2})^{\int_{N_3} e_1 \cup (\delta e_2 \cup_2 \delta e)} \times \text{bulk terms} \,, \tag{127}$$

and we extract the auxiliary degrees of freedom as

$$U_e = \prod_{e' \in N_3} \mathcal{Z}_{e'}^{\int_{N_3} e' \cup (e \cup_1 \delta e)} \prod_{e_1, e_2 \in N_3} CZ(\mathcal{Z}_{e_1}, \mathcal{Z}_{e_2})^{\int_{N_3} e_1 \cup (\delta e_2 \cup_2 \delta e)} \,, \tag{128}$$

which is the fermionic loop excitation. $U_e$ is drawn in the last of raw of Fig. 2. Notice that we have omitted the bulk part of Eq. (60) in the figures.

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
