# Peer review of "Exactly Solvable Lattice Hamiltonians and Gravitational Anomalies"

_SciPost Physics, doi:SciPost Phys. 14, 089 (2023)_

## Round 2 · Referee Report · Anonymous (Referee 1) · 2022-8-19

Strengths

1- Explicit construction of Hamiltonian model gapped boundary of beyond group cohomology SPTs. 2- The idea of integrating the Lagrange multiplier and regarding an invertible system as a finite gauge theory, then realize it as a Hamiltonian system.

Report

The manuscript describes the construction of projector hamiltonian models of gapped boundaries of SPTs that are beyond group cohomology. The idea is to realize the SPT as a finite gauge theory with a suitable DW twist, and to realize it as a Hamiltonian system. Then the gapped boundary is obtained by truncating the lattice system.
The construction as a Hamiltonian system is new and will be helpful in the study of SPT and topological phases, and thus I recommend publishing it in SciPost after the authors consider the suggestions below.

Requested changes

1- Although the construction of Hamiltonian system is new, as for a spacetime system it is discussed in https://arxiv.org/1905.05391 . The bosonic cases it is briefly discussed in Section 3. (Despite the name of the title, time-reversal symmetry is not necessarily in the construction.) It would be ideal to discuss the relationship to this construction. 2- Typo in the title of section 2.1: Beyong -> Beyond. 3- The sentence “If we denote the background..” on page 10 does not finish.

  • validity: top
  • significance: high
  • originality: high
  • clarity: top
  • formatting: excellent
  • grammar: good

Author:  Po-Shen Hsin  on 2022-08-24  [id 2749]

(in reply to Report 1 on 2022-08-19)
Category:
remark

We thank the referee for carefully reading the manuscript and pointing out the reference.

  1. The construction in Section 3 of 1905.05391 is related but different from the method in the file. There, the goal is to obtain a gapped boundary, and one way is by introducing a twisted dynamical gauge field a such that da = (Stiefel-Whitney classes). This allows one to rewrite the bulk effective action as a topological boundary term for these dynamical gauge fields a.

In our construction for the invertible phase, we introduce instead a bulk dynamical gauge field b that is constrained to satisfy b = (Stiefel-Whitney classes) . If there is a boundary, there are different boundary conditions for b. The manuscript focused on the rough or Dirichlet boundary condition, and also briefly discussed the free boundary condition. The gapped boundary in 1905.05391 corresponds to the Dirichlet boundary condition of b, which is also discussed in the manuscript.

  1. Thank you, we will fix the typo.
  2. Thank you, we will change the sentence to " Let us denote the background...".

Anonymous on 2022-08-25  [id 2750]

(in reply to Po-Shen Hsin on 2022-08-24 [id 2749])

Thank you for your comments. 1. I agree with the author's explanation, but I still think it is worth explaining in the article. 2. and 3. Thanks.

Anonymous on 2022-08-25  [id 2753]

(in reply to Anonymous Comment on 2022-08-25 [id 2750])

Thank you for the suggestion. We will add the explanation in the article.

---

## Round 2 · Referee Report · Anonymous (Referee 2) · 2022-12-22

Strengths

  1. They construct explicit lattice Hamiltonian models of (4+1)D beyond cohomology invertible phases, in the case with Z2 0-form symmetry or without any global symmetry. They also obtain the boundary theory given by Z2 gauge theory in (3+1)D.
  2. In the case without global symmetry, the boundary (3+1)D Z2 gauge theory of the beyond-cohomology invertible phase (with effective action w2w3) supports a magnetic loop excitation phrased as a 'fermonic loop' in literature. They discovered a non-trivial statistical property of the fermonic loop excitation based on the concrete lattice model and effective field theory.

Report

This paper gives concrete lattice Hamiltonian models for the beyond-cohomology invertible phases in (4+1)D. In particular, their construction of the w2w3 phase without symmetry is quite clever, where they express the w2w3 phase in terms of a higher-form Z2 gauge theory whose action does not have explicit dependence of Stiefel-Whitney classes w2, w3. This makes possible to realize a simple lattice model without employing explicit cochain representatives of w2, w3 on lattice. They also study the statistical property of the fermonic loop excitation of the (3+1)D Z2 gauge theory on its boundary, and found that its orientation-reversing defect supports a fermion.
It clarifies the existing arguments about fermionic loops in literature, and regarded as an important contribution about understanding of fermionic loops. Overall, this is an excellent paper and I can safely recommend this article for publication.

Requested changes

They provide a disentangler of the lattice model for w2w3 phase, and claim that the disentangler realizes a non-trivial quantum cellular automaton (QCA) in (4+1)D. This claim should be a conjecture, based on the other primary conjecture that the (3+1)D Z2 gauge theory w/ fermonic particle and a fermonic loop cannot be realized by a commuting projector Hamiltonian.
However, I do not know if anyone studied or even talked about this conjecture, while it is widely believed that the chiral topological phase in (2+1)D cannot be realized by a commuting projector model, which leads to a construction of non-trivial QCA in (3+1)D.
I think it would benefit community, if the author can add some comment about the conjecture about commuting projector model described above. But this is an additional comment which is not necessarily required for publication. I can already recommend this article for publication.

---

## Round 3 · List of Changes

• fixed the typos as requested in the first (earlier) referee report
  • added a paragraph on p12 before section 2.2 as requested in the first referee report, which is the same explanation in the author reply to the first report that the referee agreed upon.
  • added a paragraph on p33 as requested by the second (more recent) referee in the second report.

---

## Editorial Decision

published